# Searching for Optimal Solutions with LLMs via Bayesian Optimization

**Dhruv Agarwal**[1,†,*] **Manoj Ghuhan Arivazhagan**[2,†], **Rajarshi Das**[3,‡], **Sandesh Swamy**[2],
**Sopan Khosla**[3,‡], **Rashmi Gangadharaiah**[2]
[1]University of Massachusetts Amherst, [2]AWS AI Labs, [3]Meta AI
dagarwal@cs.umass.edu, {mghuhan,sanswamy,rgangad}@amazon.com,
{rajarshi,sopan}@meta.com

## Abstract

Scaling test-time compute to search for optimal solutions is an important step towards building generally-capable language models that can reason. Recent work, however, shows that tasks of varying complexity require distinct search strategies to solve optimally, thus making it challenging to design a one-size-fits-all approach. Prior solutions either attempt to predict task difficulty to select the optimal search strategy, often infeasible in practice, or use a static, pre-defined strategy, e.g., repeated parallel sampling or greedy sequential search, which is sub-optimal. In this work, we argue for an alternative view using the probabilistic framework of Bayesian optimization (BO), where the search strategy is adapted dynamically based on the evolving uncertainty estimates of solutions as search progresses. To this end, we introduce Bayesian-OPRO (BOPRO)—a generalization of a recent method for in-context optimization, which iteratively samples from new proposal distributions by modifying the prompt to the LLM with a subset of its previous generations selected to explore or exploit different parts of the search space. We evaluate our method on word search, molecule optimization, and a joint hypothesis+program search task using a 1-D version of the challenging Abstraction and Reasoning Corpus (1D-ARC). Our results show that BOPRO outperforms all baselines in word search ($\geq$10 points) and molecule optimization (higher quality and 17% fewer invalid molecules), but trails a best-$k$ prompting strategy in program search. Our analysis reveals that despite the ability to balance exploration and exploitation using BOPRO, failure is likely due to the inability of code representation models in distinguishing sequences with low edit-distances.

## 1 Introduction

Decision-making under uncertainty is a hallmark of intelligent behavior (Kochenderfer, 2015; Peterson, 2017), with uncertainty arising from incomplete task information (e.g., unknown constraints and preferences) (Bazerman, 2012), limited or inaccessible data due to challenging or costly collection processes (e.g., efficacy of a drug design, feedback from human users) (Benary et al., 2023), or vast solution spaces that are difficult to model comprehensively (e.g., scientific hypotheses) (Schneider, 2018; Majumder et al., 2024; 2025). Such settings, thus, require *searching* for optimal solutions— iteratively estimating the quality of plausible solutions conditioned on limited available evidence and prior beliefs to propose a candidate that is either similar to a previous observation known to be high-performing or provide new information for future decision-making, commonly referred to as the exploration-exploitation tradeoff (Cohen et al., 2007). Specifically, we are interested in answering whether a search method with large language models (LLMs) can be designed to handle this trade-off automatically.[1]

Recent work (Krishnamurthy et al., 2024) suggests that models are good at exploiting known solutions (finding local optima) but struggle to sufficiently explore the search space (finding global optima), thus, curtailing their application in real-world search tasks. Recent attempts to address this

---

*Work done during an internship at Amazon. †Corresponding authors. ‡Work done at Amazon.
[1]Our code is available at: https://github.com/amazon-science/BOPRO-ICLR-2025.

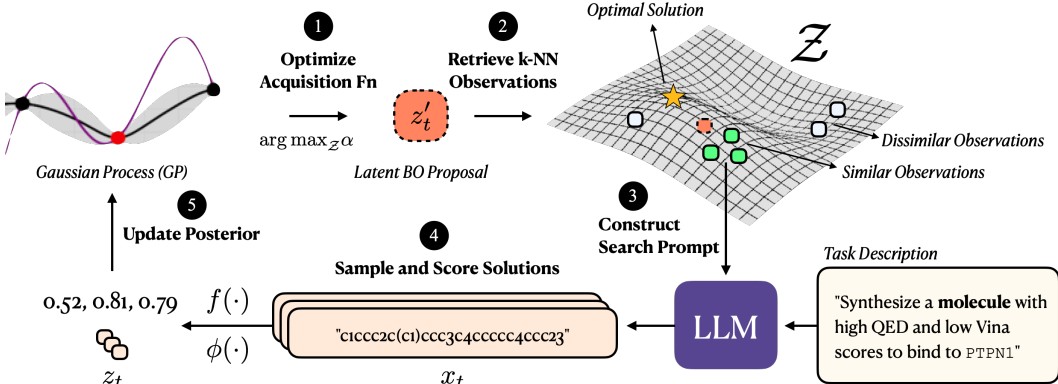

Figure 1: **Overview of Bayesian-OPRO.** In every iteration $t$, BOPRO starts by optimizing an acquisition function $\alpha$ to propose a latent vector $z'_t \in \mathcal{Z}$, the embedding space of solution sequences. $z'_t$ represents a point in the search space considered promising to observe by the surrogate model. To sample one or more solutions from $z'_t$, BOPRO constructs a search prompt for an LLM by retrieving $k$ previous observations most similar to the BO proposal in the latent space, using them as in-context examples. The sampled solutions $x_t$ are then scored using the black-box function $f(\cdot)$ and mapped to $\mathcal{Z}$ using embedding function $\phi(\cdot)$, which are then together used to update the surrogate posterior.

issue often involve constructing static meta-heuristics to encourage exploratory behavior, such as with evolutionary algorithms (Lehman et al., 2023; Meyerson et al., 2023; Romera-Paredes et al., 2024), repeated sampling (Brown et al., 2024; Yang et al., 2024), or tuning parallel-to-sequential sampling ratios based on task difficulty (Snell et al., 2024). Despite moderate success, these solutions are either too expensive, not performant enough, or require fixing a search strategy through offline evaluations. In this work, we explore an alternative direction and ask:

*Can we enable LLM-based search to automatically adapt its strategy as search progresses?*

We answer in the affirmative and propose the use of Bayesian optimization (BO) (Kushner, 1964; Frazier, 2018), a framework that builds a surrogate of an objective function using a probabilistic model initialized with a set of priors and initial samples, then generates a posterior predictive distribution over candidate solutions by iteratively integrating new evidence. While there has been preliminary work in trying to combine LLMs with BO, prior works have either viewed the task as a selection problem from an existing pool of candidates (Kristiadi et al., 2024; Opsahl-Ong et al., 2024) or relied on heuristics over samples from the LLM itself to build unprincipled uncertainty estimates (Liu et al., 2024).

We, therefore, propose **Bayesian-OPRO** (BOPRO)—a novel method for integrating BO with LLMs that leverages the evolving uncertainty in estimates of the solution space to *generate* promising solutions in each round of search to find the global optima. In each iteration, BOPRO uses latent-space BO to propose a latent target for the LLM, which is used to sample solutions via a prompting strategy that builds upon a recent in-context optimization method called OPRO (Yang et al., 2024). Specifically, we prompt the LLM with previously generated solutions that are most *similar* to the BO proposal vector in order to bias its generation towards the BO-proposed region in the search space. We empirically validate our approach on three challenging language-based search tasks— word search, molecule optimization, and hypothesis+program search—and find that BOPRO outperforms the baselines on all tasks except program search. Notably on Semantle, while baselines plateau, likely getting stuck in local optima, BOPRO shows steady gains, pointing to its ability to adaptively explore and exploit the search space. We analyze the search behaviors of each method and demonstrate that BOPRO indeed shows the capacity to balance exploration and exploitation, which is important for complex reasoning tasks. Despite this ability, we find evidence to suggest that failure on program search likely occurs due to an unavailability of fine-grained code representations that can distinguish between sequences with low edit-distances.

In summary, our contributions are: **(1)** we propose BOPRO (§5), a novel method for integrating BO with LLMs that adapts to the problem difficulty and evolving uncertainties as search progresses. **(2)**

We conduct a thorough empirical evaluation (§6) on three search tasks, validating the efficacy of BOPRO in finding optimized solutions compared to several strong baselines (§7). **(3)** We highlight program synthesis as a failure case for BOPRO and present detailed analyses that (a) clearly demonstrate a strong exploration capability in BOPRO (§8.1), and (b) reveal that the lack of fine-grained representations is likely the root cause of optimization failure (§8.2).

## 2 SEARCH TASKS

We first describe three language-based search tasks that we experiment with in this work in order to provide grounding for subsequent sections.

### 2.1 SEMANTLE: WORD SEARCH

**Semantle**[2] is a synthetic word-search task, where the goal is to find a held-out English word (e.g., *"polyethylene"*) with a limited number of guesses (budget) given only scalar feedback scores that describe the similarity of a guessed word to the held-out word. In particular, we use representations from the SimCSE (Gao et al., 2021) embedding model as the black-box scoring function. Each held-out word is also paired with an initial *warm-start* set of words and scores (e.g., *"hope" (score: 0.29), "golfing" (score: 0.35), "serum" (score: 0.51)*), which together define a problem instance.

### 2.2 DOCKSTRING: MOLECULE OPTIMIZATION

**Dockstring** (García-Ortegón et al., 2022) provides a benchmark of challenging molecule optimization tasks that are closely related to real problems in drug discovery. In this work, we focus on the multi-objective task of synthesizing a drug molecule (using SMILES strings (Weininger, 1988)) that jointly optimizes for both druglikeness, as measured by QED scores (Bickerton et al., 2012), and binding (or docking) affinity to a target protein site, measured by Vina scores (Trott & Olson, 2010). Following Yuksekgonul et al. (2024), we synthesize new molecules for each protein target starting from 3 basic chemical fragments—benzene (`c1ccccc1`), pentane (`CCCCC`), and acetamide (`CC(=O)N`).

### 2.3 1D-ARC: HYPOTHESIS+PROGRAM SEARCH

The **Abstraction and Reasoning Corpus** (Chollet, 2019) is a set of analogy puzzles in which a solver must infer the common abstract rule underlying a small set of 3-5 "train" grid transformations, and apply that rule to a new "test" grid. In this work, we use a 1-dimensional version of the dataset, **1D-ARC**, introduced by Xu et al. (2024).[3] We take an inductive hypothesis+program search view (Wang et al., 2024), wherein search is performed over the space of natural-language algorithms followed by generating its implementation as a python program. The goal for each problem instance is, therefore, to find an algorithm and its python implementation that can solve all the train examples, followed by a single evaluation of the best found program on a test example to receive a score.

## 3 BACKGROUND: BAYESIAN OPTIMIZATION

Bayesian optimization (BO) is a framework for derivative-free, global optimization to find

$$x^* := \arg\max_{x \in A} f(x),$$

where $f : \mathbb{R}^d \to \mathbb{R}^o$ is the black-box function to optimize, $A$ is a feasible set within the input space (often defined by box constraints) to constrain search, $d$ is the number of input dimensions to tune, $o$ is typically the number of scalar objectives used to evaluate $f$ (we set $o = 1$ throughout this work), and we are allowed a limited budget $T$ of evaluations that can be made using the black-box function

---

[2]Inspired by the online word game of the same name: `https://semantle.com/`.

[3]We use the 1D version to remove the need for identifying non-sequential object cohesion in 2D grids, noted as being challenging for both moderate-sized models and uni-modal text models (Xu et al., 2024).

$f$ to get an observation of the objective at some point in the input space. The goal for BO is then to iteratively select a point for evaluation that is most likely to be $x^*$ based on the available evidence.

In a typical BO implementation, we first initialize a Bayesian statistical model, called a **surrogate**, to model $f$ with either user-defined priors about the objective function or with uninformed priors. Typical choices for the surrogate model include Gaussian processes (GPs) (Rasmussen & Williams, 2006) and Bayesian neural networks (BNNs) (Springenberg et al., 2016), which encode distributions over plausible realizations of the true black-box function. Once initialized, a set of **warm-start experiments** are then conducted by taking random samples from the input space and observing their black-box values. These samples are used to compute the initial **posterior predictive distribution** $f(x) \mid f(x_{1:t-1})$, which, in the case of a GP, is a normal distribution with mean $\mu_t(x)$ and variance $\sigma_t^2(x)$ at each input $x$ at time-step $t$. The final component is an **acquisition function** $\alpha : \mathbb{R}^d \to \mathbb{R}$, which is a function that balances exploration and exploitation by assigning a scalar value to each point in the input space denoting the utility of sampling that point based on the posterior predictive at a given time-step. At each step $t$, the next point is then sampled as

$$x_t := \underset{x \in A}{\arg\max}\, \alpha(x, f(x) \mid f(x_{1:t-1})),$$

which can be optimized using standard gradient-based methods. We then evaluate the black-box function at this new point to get $f(x_t)$ and update the posterior predictive. This loop is repeated until either the search budget is exhausted or a solution with sufficient quality is found, following which the best solution found thus far is returned.

## 4 RELATED WORK

**Test-time compute with LLMs.**  Snell et al. (2024) provide a framework for categorizing different methods for scaling test-time compute into refining the proposal distribution and using verifiers to guide decoding, e.g. during beam search. Brown et al. (2024) present an analysis of the ability of repeated sampling to scale test-time compute and derive scaling laws for the same. Several other test-time search and optimization methods (Yang et al., 2024; Lehman et al., 2023; Meyerson et al., 2023; Romera-Paredes et al., 2024; Wang et al., 2024) approach the problem assuming the availability of a known verifier, similar to our setting.

**Bayesian optimization and LLMs.**  Some methods use the LLM itself to simulate each step of BO via repeated prompting to compute uncertainties over candidate solutions (Liu et al., 2024; Ramos et al., 2023). Other methods have used BO with LLMs to perform efficient *selection* over a discrete solution set (Kristiadi et al., 2024; Opsahl-Ong et al., 2024). Chen et al. (2024) propose an approach to learn soft-prompts with BO starting with a set of random vectors and show promising results on instruction induction. Finally, our work is also related to earlier works performing LSBO with autoencoders instead of LLMs (Gómez-Bombarelli et al., 2018).

## 5 BAYESIAN-OPRO (BOPRO)

We now introduce BOPRO (Fig. 1)—a method for LLM-based search that can automatically adapt its strategy based on the evolving uncertainties about the search space using latent-space BO over textual embeddings of plausible solutions for a given task. In each iteration with BOPRO, a surrogate model builds a probabilistic estimate of task performance for unobserved regions of the search space using previous observations and proposes a vector representing a promising region to sample solutions from next. The proposal vector is used to build a *search prompt* that is input to an LLM to sample new solution sequences. The generated solutions are then scored using the black-box function for the task and added to the pool of observations to improve surrogate estimates in subsequent iterations. This process is repeated until either the optimal solution is found or the search budget is exhausted, following which the best solution found thus far is returned.[4]

---

[4]Our focus is to evaluate search methods and, thus, we use tasks with external verifiers to avoid confounding with noisy feedback. However, our method can easily be used with learned verifiers such as reward models.

We next describe each component of BOPRO categorized broadly into: (1) generating the latent-space proposal using BO, and (2) decoding the solution text from the latent proposal via prompting.

## 5.1 LATENT-SPACE BO

### 5.1.1 WARM-STARTING

To generate the initial warm-start set $\{(x_w^{(i)}, f(x_w^{(i)}))\}_{i=1:W}$, we prompt an LLM to generate $W$ unique solutions using a manually written task description (see Appendix A.6.1). We then compute the latent-space embedding for each generated solution using a representation function $\phi(\cdot)$, e.g., a text embedding model, to get $z := \phi(x) \in \mathbb{R}^d$. We then derive the initial posterior predictive distribution conditioned on $\{(z_w^{(i)}, f(x_w^{(i)}))\}_{i=1:W}$. Note that the goal at this stage is only to provide wide coverage of the solution space, and not to maximize $f$.

### 5.1.2 GENERATING REPRESENTATIONS

**Representation prompt.** To generate useful representations for solution candidates, we convert the raw text of a solution into a task-dependent prompt before passing it through $\phi(\cdot)$. E.g., in program search, the algorithm $x$ and code $y$ may simply be represented as "`Algorithm: x; Code: y`" (see Appendix A.6.2).

**Representation function.** BOPRO relies on the availability of an embedding function $\phi(\cdot)$ that is capable of generating good representations of plausible solutions for the task. For example, in program search, embeddings must be able to distinguish between distinct algorithmic constructs. Additionally, since it is widely reported that GPs struggle with large input dimensions (Snoek et al., 2012; Wang et al., 2013), we, optionally, include a dimensionality reduction function $\psi(\cdot)$ to get $z := \psi(\phi(x)) \in \mathbb{R}^{d'}$, where $d' \ll d$. Several choices exist for $\psi$, including PCA, low-rank projection (Li et al., 2018; Aghajanyan et al., 2021), and random projection (Wang et al., 2013). When no dimensionality reduction is used, $\psi$ is simply set as the identity function.[5]

### 5.1.3 GENERATING LATENT-SPACE PROPOSALS

**Surrogate model.** BOPRO uses a probabilistic surrogate model to estimate the uncertainty in the search space and efficiently update its beliefs about the goodness of different regions as new solutions are observed. Throughout this work, we use a Gaussian process (GP) surrogate model with a Matérn kernel (Genton, 2001) and priors manually-tuned by visually assessing the fit of the posterior distribution on a known set of solution-score pairs for each task on a randomly sampled problem instance (see Fig. 7).

**Acquisition function.** As described in §3, given a posterior predictive distribution over the feasible search space, BO uses an acquisition function to estimate the *value* of observing each point. We optimize this acquisition function using multi-start projected gradient ascent to get a new proposal vector that represents a promising region of the search space to sample solutions from. We experiment with three common choices: log expected improvement (**LogEI**) (Ament et al., 2023), upper-confidence bound (**UCB**) (Auer, 2002), and Thompson sampling (**TS**) Thompson (1933).

Please see Appendix A.4.2 for additional details on our BO setup.

## 5.2 LATENT-TO-TEXT DECODING

With a latent BO proposal that represents a promising region of the search space, we now require a mechanism to sample textual sequences from the proposed region.

---

[5]We found that using no dimensionality reduction results in the best performance and is, hence, our default setting for all experiments. Please see Appendix A.5.6 for an ablation result.

### 5.2.1 GENERATING SOLUTIONS VIA SEARCH PROMPTS

Given a BO proposal vector $z'_t$, we first compute the cosine similarity $s_{z'_t, z}$ between $z'_t$ and each previously observed solution vector (including the warm-start set) $z \in Z := \{z_w^{(i)}\}_{i=1:W} \cup \{z_j\}_{j=1:t-1}$. We then use the resultant solution-score pairs to construct a **search prompt** by selecting and ordering the top-$k$ solutions according to each $s_{z'_t, z}$ in order to provide an inductive bias for generating solution $x_t$ such that $z_t := \psi(\phi(x_t))$ has higher similarity to $z'_t$ than previous solution vectors $z \in Z$ (see Appendix A.6.3). In practice, we sample a batch of candidates for each $z'_t$, which we found improved performance for all methods we evaluated.[6]

**Generalizing OPRO.** Our method is a Bayesian generalization of a recent in-context optimizaton method, OPRO (Yang et al., 2024), which instead constructs a search prompt by selecting the top-$k$ observed solutions according to the black-box function itself, thereby using a greedy strategy. As an example, consider a Semantle problem with the hidden word being *"wordsmith"* and the following previous guesses: *"word"* (score: 0.7), *"sentence"* (score: 0.65), *"lock"* (score: 0.5), and *"metal"* (score: 0.49). Using OPRO, we would greedily select *"word"* and *"sentence"* to include in the prompt, whereas BOPRO could sample *"lock"* and *"metal"* based on the evolving uncertainties in the BO process, thus possibly generating *"locksmith"*, which can further generate *"wordsmith"*. Different from OPRO, we also find that removing numerical scores from the prompt results in a modest improvement across methods, therefore we use this as the default setting.

**Alternative decoding strategies.** Note that our generalization of OPRO can similarly be extended to other in-context optimization methods, e.g., LMX (Meyerson et al., 2023) and FunSearch (Romera-Paredes et al., 2024), which are LLM-based evolutionary algorithms. To demonstrate this generalization, we also show experimental results with one such extension (Bayesian-LMX), but find that it consistently underperforms BOPRO.

**Handling sampling errors.** There are two types of errors that can occur when repeatedly generating solutions from an LLM. First, a generated solution may have already been sampled in a previous iteration, and second, a generated solution might be invalid with respect to the black-box function, e.g., a generated python program may have syntax errors. To address both problems, we follow prior works and leverage retry ("self-refine") loops. Please see Appendix A.4 for additional details.

## 6 EXPERIMENTS

To validate the search performance of BOPRO, we run a comprehensive set of experiments on three challenging language-based search tasks as introduced in §2—**Semantle** (word search), **Dockstring** (molecule optimization), and **1D-ARC** (hypothesis+program search).

### 6.1 BASELINES

We evaluate three variants of BOPRO characterized by the acquisition function used: **BOPRO-logEI**, **BOPRO-UCB**, and **BOPRO-TS**. Our main baseline is **OPRO (Greedy)** (Yang et al., 2024), which replaces the Bayesian prompting in our method with a greedy best-$k$ strategy. We also compare with a **repeated sampling (RS)** baseline[7], which relies on the stochasticity from ancestral sampling to generate new solutions from a fixed initial prompt. Lastly, we include a **random** baseline that samples $k$ previously observed solution-score pairs in each iteration to include in the prompt.

**Additional baselines (§7.4).** We also evaluate **InstructZero** (Chen et al., 2024), an approach that replaces the hard-prompting in BOPRO with a soft-prompt that is iteratively optimized using BO. Additionally, as described in §5, it is straightforward to generalize our Bayesian view to other in-context optimization methods, including evolutionary algorithms, and we present an experiment with one such method, **LMX** (Meyerson et al., 2023).

---

[6] While every iteration in BOPRO may not yield $z_t$ that is close to $z'_t$, we show in experiments that the generations are sufficient to guide search towards optimized solutions over time.

[7] Note that in Semantle, the initial warm-start is part of the task description, so we make it available as a fixed prompt to RS. For 1D-ARC, we manually write a common task description used across all methods.

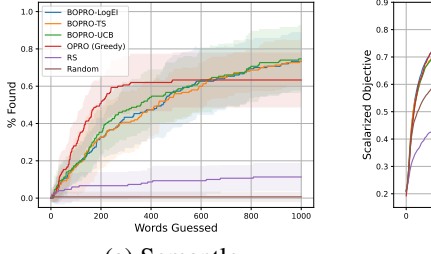
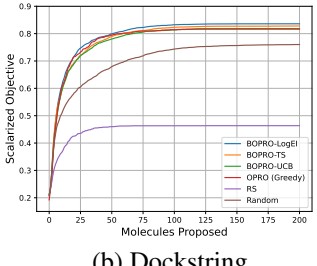
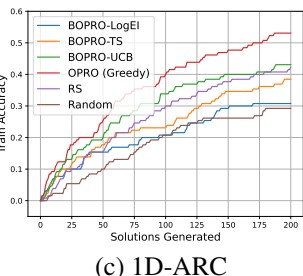

(a) Semantle        (b) Dockstring        (c) 1D-ARC

Figure 2: **Best-so-far search performance (using Mistral-Large). (a)** On Semantle, all BOPRO variants outperform greedy OPRO in finding the hidden word by $\geq$10 percentage points. **(b)** In molecule optimization, BOPRO shows slightly better performance than OPRO on average. Importantly, OPRO is only able to finish optimization for 12 of 58 protein targets in the same wall-clock time as BOPRO due to consistently proposing invalid sequences (causing high latency in black-box evaluation). **(c)** In program search, BOPRO trails OPRO due to code embeddings that are unable to distinguish between sequences with low edit-distances (see §8.2). **Takeaway:** BOPRO generates higher-quality solutions than all baselines on the tasks of word search and molecule optimization, but shows a failure scenario with hypothesis+program search.

## 6.2 MODELS

Our main results use Mistral-Large-2407 (Jiang et al., 2023) as the representative LLM to generate solutions for each search method. We also present experiments with additional model families such as LLaMA-3.1-8b-Instruct (Dubey et al., 2024), Gemma-2-2b-It (Team et al., 2024), and GPT-4o (Achiam et al., 2023). For the embedding model, on the tasks of Semantle and 1D-ARC, we use GTE-Qwen-2-1.5b-Instruct (Li et al., 2023), an instruction-tuned embedding model, since it is pretrained on both general-purpose English data as well as code, both of which are relevant to the tasks in question. For molecule optimization (Dockstring), we use the Molformer (Ross et al., 2022) embedding model, which was jointly pre-trained on natural-language text and SMILES strings.

Please see Appendix A.4 for additional details of our experimental setting.

## 7 MAIN RESULTS

### 7.1 SEMANTLE: WORD SEARCH

In Fig. 2(a), we show the main results on Semantle by describing the percentage of problems where the hidden word was found over 50 problem instances (using 10 hidden words and 5 warm-start sets) with a maximum budget in each run set to 1000 guesses and averaged over 3 repeat runs.

**Results.** First, we observe that both repeated sampling and the random baseline fail to find the hidden word in most runs, thus validating that Semantle is a challenging test bed for search that requires a substantial shift in the sampling distribution as more evidence is gathered. Next, we find that all BOPRO variants outperform the baselines by $\geq$**10 percentage points**. Notably, while OPRO shows rapid initial gains, it is followed by a long plateau, likely indicating that it gets stuck in local optima. On the other hand, BOPRO shows steady gains, pointing to its capability to adaptively explore and exploit the search space.

### 7.2 DOCKSTRING: MOLECULE OPTIMIZATION

Fig. 2(b) shows our results for molecule optimization to synthesize optimized drug molecules for 58 clinically relevant protein targets from the Dockstring (García-Ortegón et al., 2022) benchmark. In order to perform multi-objective optimization over both druglikeness (QED) and binding affinity (Vina scores), we follow the common practice in BBO and maximize instead a single scalarized metric that takes a weighted combination of the two scores (see Appendix A.4.1). For each protein

target, we start optimization with a warm-start set of benzene (`c1ccccc1`), pentane (`CCCCC`), and acetamide (`CC(=O)N`), and generate a maximum of 200 new molecule SMILES strings.

**Results.** Our results show that BOPRO marginally outperforms greedy OPRO on average. More notably, we find that OPRO is unable to even finish optimization for all 58 targets (completing only 12) over a 2-day period, which is the same wall-clock time allocated to BOPRO. Analyzing the generations, we find that the greedy approach in OPRO consistently proposes SMILES strings that are $40\%$ longer than those proposed by BOPRO. This increases evaluation time with the black-box function and, in fact, results in $17\%$ more invalid molecules being proposed than with BOPRO. In Fig. 3, we also visualize the distribution of druglikeness and affinity scores of

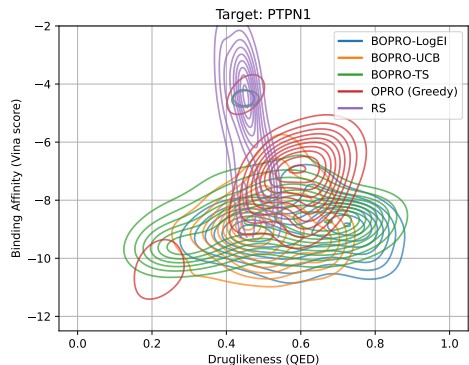

Figure 3: SMILES distributions for PTPN1

molecules generated by each method for one protein target, tyrosine-protein phosphatase non-receptor type 1 (PTPN1). We see that OPRO is unable to jointly optimize both objectives but all BOPRO variants show an ability to explore the complex manifold and propose molecules that have high druglikeness and better binding affinity (lower Vina score).

### 7.3 1D-ARC: HYPOTHESIS+PROGRAM SEARCH

**Constructing 1D-ARC-Hard.** Following Wang et al. (2024), we run a preliminary evaluation on their filtered subset of 108 problem instances. However, our results show that with only simple repeated sampling (RS) for 100 iterations using Mistral-Large, it is possible to achieve a test set accuracy of $81.48\%$, 8 points higher than the method in Wang et al. (2024) that uses GPT-4. We, therefore, construct **1D-ARC-Hard**, a challenging subset suitable for evaluating search performance by first running RS for 100 generations for each of the 901 problems in the original dataset, followed by sampling 130 instances from the unsolved subset of 175 problems.

**Results.** Our results are shown in Fig. 2(c) and Fig. 4. Unlike previously, we find that all BOPRO variants now trail the greedy OPRO baseline, with BOPRO even underperforming repeated sampling, noted as being a strong method for code generation in prior work (Brown et al., 2024). We analyze this failure scenario and present our hypothesis for the root cause in §8.

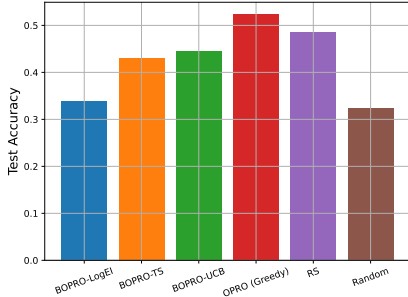

Figure 4: 1D-ARC test accuracy

### 7.4 ADDITIONAL EXPERIMENTS

Below, we present experiments with additional baselines that represent two distinct LLM-based iterative optimization methodologies.

| Method | Score |
|---|---|
| InstructZero | 4.00% |
| InstructZero (w/ sampling) | 8.00% |
| RS | 14.00% |
| BOPRO-LogEI | 66.67% |
| BOPRO-UCB | 76.67% |
| BOPRO-TS | 73.33% |

Table 1: InstructZero on Semantle

**InstructZero.** InstructZero is a method that replaces hard-prompts, as in BOPRO, with soft-prompts that are also optimized using BO to steer LLM generations towards higher utility. We evaluate this method on all 50 problem instances in Semantle using Llama-3.1-8b-Instruct and compare it with BOPRO in Table 1. We use both the original formulation of InstructZero as well as a version that uses ancestral sampling instead of greedy decoding. Our results show that InstructZero is unable to outperform even repeated sampling, indicating the lack of any meaningful optimization.

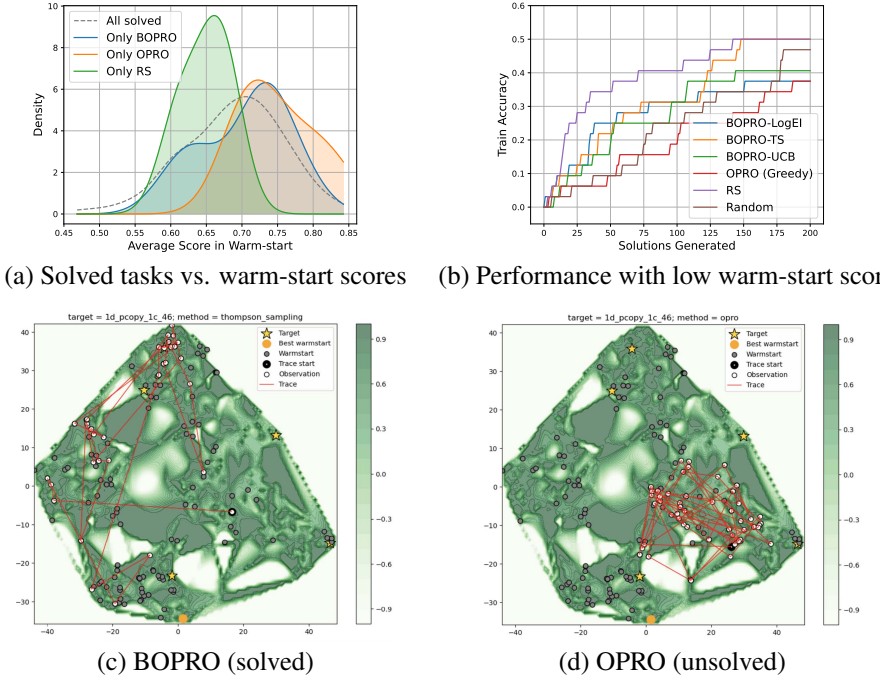

(a) Solved tasks vs. warm-start scores      (b) Performance with low warm-start scores

(c) BOPRO (solved)            (d) OPRO (unsolved)

Figure 5: **Exploration-exploitation trade-off with BOPRO. (a)** We show the distribution of tasks exclusively solved by each method versus the average warm-start scores for those tasks. OPRO is able to solve tasks with high-scores showing an exploitation strategy, random sampling solves tasks with low scores due to no dependence on prior generations (thus, using an exploration-only strategy), and BO shows a bimodal distribution solving both low- and high-scoring tasks, indicating adaptability to task difficulty. **(b)** On tasks with low warm-start scores, BOPRO and other baselines outperform greedy. **(c, d)** Search trajectories for 1 problem instance of 1D-ARC; OPRO gets stuck in a local optima, while BOPRO-TS is able to explore different regions and find the correct solution.

**LMX.** Next, we evaluate the LMX evolutionary algorithm and also present its Bayesian generalization, **BLMX**, as described in §5. Our results on 1D-ARC show that both LMX and BLMX are viable alternatives to OPRO and BOPRO, respectively, but often trail in performance. Please see Appendix A.5.2 for more details.

**Generalization to other LLMs.** Finally, we also validate our method with other LLMs (of varied sizes) by running additional experiments on Semantle with GPT-4o and Gemma-2-2b-It. Our results are shown in Appendix A.5.1 and demonstrate similar trends to our main results.

## 8 WHY IS BOPRO UNDERPERFORMING ON PROGRAM SEARCH?

To determine the root cause of failure of BOPRO on program search, we next present a detailed analysis of our 1D-ARC results.

### 8.1 IS BOPRO UNABLE TO BALANCE EXPLORATION-EXPLOITATION?

To answer this question, we design an experiment starting search with different amounts of information available in the warm-start set. Arguably, tasks with low warm-start scores are more challenging and require more exploration, whereas tasks with high warm-start scores can more easily be solved by greedily exploiting the best observed solution. In Figure 5(a), we plot the probability density of problem instances solved by each method as a function of the average scores observed in the warm-start set. We find that OPRO performs better on tasks with higher warm-start scores, suggesting an exploitation strategy. With BOPRO, on the other hand, we observe a bimodal distribution that covers both low- and high-scores, indicating the ability to balance exploration and exploitation.

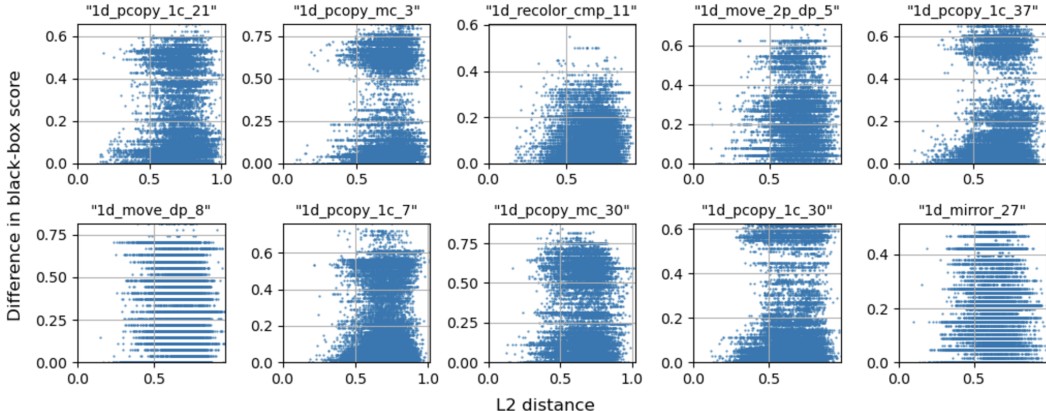

Figure 6: **Diagnostic for representation quality.** The scatter plot shows the distance between candidate solutions of 10 problems in 1D-ARC in the representation space and the corresponding difference in terms of their black-box scores. Useful representations for BO should result in no points in the top-left quadrant of these plots.

We verify these observations by plotting performance on a subset of 1D-ARC-Hard tasks that have low warm-start scores (below the first quartile). As shown in Fig. 5(b), OPRO suffers a sharp drop in performance whereas BOPRO, RS, and even random show improvements, with BOPRO-TS and RS showing the best performance. We also provide visual evidence in Fig. 5(c,d) for demonstrating the ability of BOPRO in balancing exploration-exploitation by plotting the search trajectories in the latent search space (see Appendix A.5.3 for more details). **Takeaway:** BOPRO does show the ability to balance exploration-exploitation and is, thus, not the cause of failure in program search.

## 8.2 ARE OFF-THE-SHELF CODE EMBEDDINGS ILL-SUITED FOR PROGRAM SEARCH?

Since BOPRO relies on good representations of the solution space in order to perform meaningful optimization, a likely cause of failure is poor representation quality from off-the-shelf code embedders, which may not provide fine-grained signal for optimizing code sequences. To verify this hypothesis for 1D-ARC and the GTE-Qwen embeddings, we present a diagnostic scatter plot (Korovina et al., 2020) in Fig. 6 that shows pairwise L2 distances between solution candidates as well as their corresponding differences in black-box scores[8]. Ideally, when L2 distance is low, the score difference should also be low in order to provide useful signal to the nonparametric GP surrogate. From our plot, however, we clearly see that this is not the case, i.e., solutions deemed similar in the embedding space vary dramatically in terms of task scores. This is intuitive as well—consider two python code sequences that only differ in one inequality operator. Off-the-shelf embeddings do not capture this nuanced difference, but such a change is likely to have an impact on program output. **Takeaway:** low edit-distances between candidate solutions and the unavailability of fine-grained code embedding models are a likely cause of failure for BOPRO in program search, indicating an important direction for future work.[9]

## 9 CONCLUSION

We propose a novel methodology, Bayesian-OPRO (BOPRO), for LLM-based search using Bayesian optimization, which uses the evolving uncertainty in estimates of the search space to propose promising new solutions in each iteration to find the global optima. Our experiments on word search and molecule optimization show that BOPRO outperforms strong baselines. Notably, BOPRO outperforms OPRO by $\geq 10$ percentage points on Semantle and produces 17% fewer invalid molecules on Dockstring. On the challenging hypothesis+program search task of 1D-ARC, however, BOPRO trails OPRO. We investigate this failure and find that despite BOPRO showing the ability to balance exploration and exploitation, the problem lies in the inadequacy of off-the-shelf code embedding models in providing distinct representations for sequences with low-edit distances.

---

[8]See Fig. 11 for the same plot on Semantle.

[9]See Appendix A.1 for an additional discussion on limitations and future work.

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

## A   APPENDIX

### A.1   LIMITATIONS AND FUTURE WORK

There remain several questions that lie beyond the scope of our immediate investigation but would be useful to consider in future work. For instance, how can we accurately bound the space of valid solutions in the latent embedding space in order for acquisition optimization to always give us a valid latent proposal as well as not artificially constrain it to a limited region. Currently, we define the bounds for acquisition optimization as $1.5\times$ the convex hull defined by the observed warm-start set. However, this may either be too large or too small, depending on the problem. Another unanswered question is whether BO is negatively affected by the existence of multiple global optima, such as in 1D-ARC. Lastly, it is evident that using a prompting-based decoding method to translate latent proposals back to discrete textual tokens does not offer precise control. Future work should investigate methods that can improve the accuracy of converting latent proposals to text.

### A.2   ADDITIONAL METHOD DETAILS

#### A.2.1   HANDLING SAMPLING ERRORS

**Repetitions.**   Since our method repeatedly samples an LLM to generate new solutions, generation of duplicate samples is common. We, therefore, use a limited number of re-sampling calls to the LLM to try and generate new solutions in each round of search. If all calls result in repeated solutions, we accept the generation but use the BO proposal as the representation of the generated sequence along with the observed black-box value to update the surrogate in the next iteration.[10]

**Invalid generations.**   In tasks such as code generation, some proposed generations may be invalid, e.g., a generated program may have syntax errors. In such cases, we use a limited number of self-refine (Madaan et al., 2024) calls that attempt to fix the generated solution using the same LLM. If errors persist, we assign the solution a score of $-1$, assuming a $[0, 1]$ range for valid scores.

#### A.2.2   EXTENDED BO DESCRIPTIONS

**Acquisition functions.**   We evaluate BOPRO with three types of acquisition functions that each uses the posterior distribution differently to estimate the value of sampling the next point.

- **Log expected improvement (LogEI)** computes the expected gain in performance for each point with respect to the best known solution so far.

---

[10]We found this technique reduced repetitions in our experiments. This is likely due to noisy updates allowing the acquisition optimization to escape local optima.

- **Upper-confidence bound (UCB)** computes the value as the sum of the mean value of a point and the standard deviation (uncertainty) at the point weighted by an exploration factor.
- **Thompson sampling (TS)** randomly samples a function from the posterior distribution and uses the estimates from it as the acquisition values.

**Acquisition optimization bounds.** To optimize the acquisition function and generate a new BO proposal, we use multi-start projected gradient-ascent, which requires the specification of bounds to constrain the proposals. We use a heuristic of setting each dimension of the input embedding space to 1.5x the range observed in an *a priori* collection of unlabeled candidate solutions. When such a collection is unavailable, we make use of the warm-start set instead.

**Tuning priors.** We sample a set of known solution-score pairs, order them in ascending order of their black-box scores, and plot the mean and standard deviation of the posterior distribution for each round of BO to see if the posterior progressively fits to the ground-truth function as shown in Fig. 7.

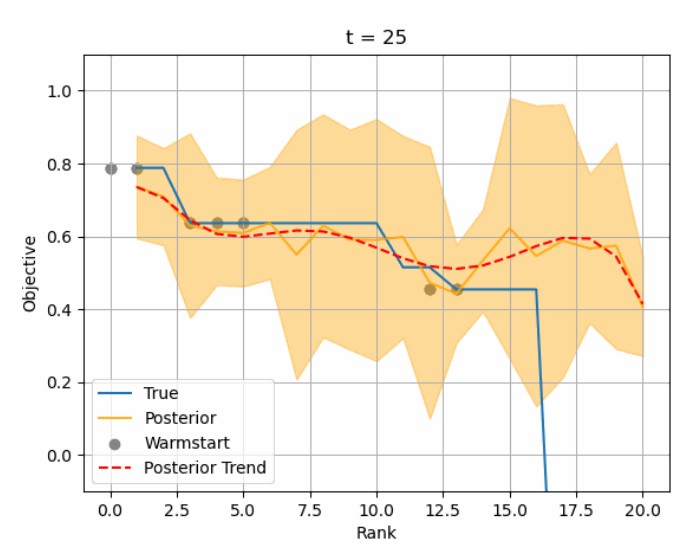

Figure 7: Prior tuning example on 1D-ARC.

## A.3 ADDITIONAL TASK/DATASET DETAILS

### A.3.1 ARC AND 1D-ARC EXAMPLES

In Fig. 8, we show two examples of the type of grid transformations that are required in the ARC and 1D-ARC datasets.

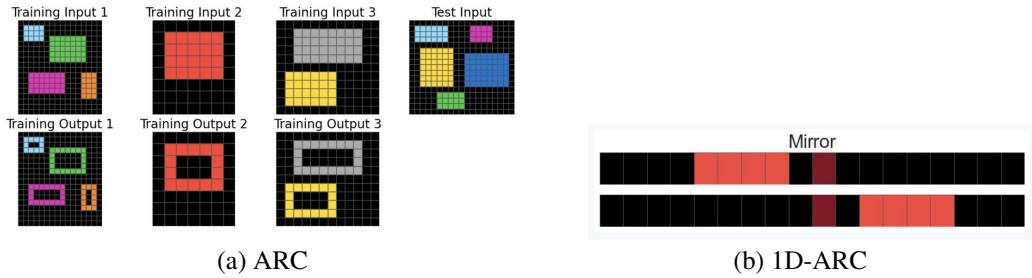

(a) ARC        (b) 1D-ARC

Figure 8: ARC (Chollet, 2019) and 1D-ARC (Xu et al., 2024) example tasks.

### A.4 EXPERIMENTAL SETTINGS

#### A.4.1 TASK-SPECIFIC SETTINGS

**Semantle.** The black-box function we use is the cosine similarity of vector representations generated using the SimCSE (Gao et al., 2021) sentence embedding model, where the score for a candidate word $x$ for a hidden word $y$ is computed by comparing sequences "What is a {x}?" and "What is a {y}?". The number of warm-start candidates is 20, number of solution generations is 1000, BO optimization batch size is 1 (i.e., in each iteration we only sample one proposal vector by optimizing the acquisition function), BO decoding batch size is 10. The total BO iterations are, therefore, 100. We also use repeat retries with a maximum count of 3. During decoding, we use 10 in-context examples. The representation model we use is GTE-Qwen2-1.5b-Instruct.

**Dockstring.** To perform multi-objective optimization over both druglikeness (QED scores) and binding affinity (Vina scores), we scalarize the objective into a single metric. First, we normalize vina scores by using their typical range of -3 to -12. We then take its negative and compute a weighted combination of this negative normalized Vina (NNV) score and QED scores as: $0.2 \times \mathrm{QED} + 0.8 \times \mathrm{NNV}$. Given a new SMILES string generated by the LLM, we follow the mechanism in Yuksekgonul et al. (2024) to compute QED and Vina scores. Number of warm-start candidates is 3, number of solution generations is 200, BO optimization batch size is 1, BO decoding batch size is 10, number of in-context examples is 10, number of repeat retries is 3, and the representation model is Molformer.

**1D-ARC.** Since the dataset imposes only a 0-1 accuracy as the objective, we introduce a **Hamming-distance based metric** to provide process-level supervision for different search methods. Specifically, we calculate the objective as the proportion of cells in the predicted grid that match the ground-truth grid. If there is a shape mismatch, we assign a score of -1. See Appendix A.3.1 for an example task. Number of warm-start candidates is 100, number of solution generations is 200, BO optimization batch size is 1, BO decoding batch size is 10, number of in-context examples is 3, number of unique retries is 0, the number of self-refine trials for code fixes is 1, and the representation model is GTE-Qwen-2-1.5b-instruct.

#### A.4.2 BO SETUP

We use BoTorch (Balandat et al., 2020) and GPyTorch (Gardner et al., 2018) for all BO-related code in our experiments. We use output standardization and no input normalization, but our embeddings generated using Sentence-Transformers are already normalized. We use a GP as the surrogate model and manually tune the lengthscale and outputscale priors for a Matérn 5/2 kernel (Genton, 2001) by visualizing whether the posterior mean and standard deviation conform to the true black-box function in successive BO iterations over a random sample of known set of solution-score pairs for a single instance of a task (see Figure 7). Specifically, we use a gamma prior with concentration=4 and rate=2 for the lengthscale (setting ard_num_dims=1) and outputscale, and a normal prior with mean=0.4 and std=0.01 for the mean function. We also add noise variance of 0.001.

#### A.4.3 LLM AND EMBEDDINGS SETUP

We use Huggingface transformers (Wolf, 2019) library and AWS Bedrock APIs to access the LLMs used in this work. For embeddings, we use the sentence-transformers (Reimers & Gurevych, 2019) library along with models hosted on Huggingface. The decoding parameters used for sampling solutions from the LLM are temperature=1.0, top_p=0.9, max_new_tokens=512.

### A.5 ADDITIONAL EXPERIMENTS AND RESULTS

#### A.5.1 PERFORMANCE WITH ADDITIONAL LLMS ON SEMANTLE

In Fig. 9, we show experiments on Semantle using two additional LLMs on opposite ends of the scale spectrum—Gemma-2-2b-It and GTP-4o.

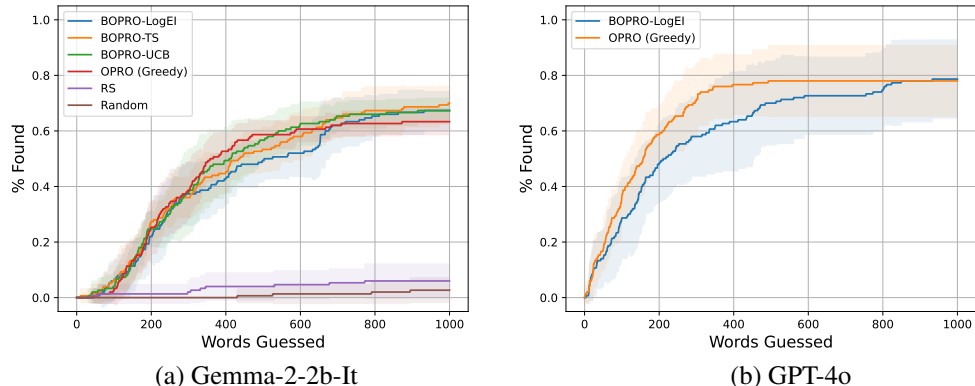

(a) Gemma-2-2b-It
(b) GPT-4o

Figure 9: **Performance on 50 Semantle problem instances. (a)** Using Gemma-2, we see the same trend as with Mistral-Large of BOPRO outperforming OPRO, albeit with a smaller gap. **(b)** Using GPT-4o, BOPRO only marginally outperforms OPRO, but demonstrates a similar tendency of OPRO in showing rapid initial gains followed by a plateau versus the behavior of BOPRO, which shows steady gains. Note that we able to evaluate on two methods due to budget constraints.

### A.5.2 PERFORMANCE WITH LMX AND BLMX ON 1D-ARC

The BOPRO methodology can similarly be extended to other in-context optimization methods by replacing the objective scores in constructing prompts with latent-space similarity scores to the BO proposal vector. For instance, below is a description for Bayesian-LMX (BLMX).

**Bayesian-LMX.** LMX ("language model cross-over") is an evolutionary algorithm that iteratively builds a new generation of child solutions by prompting the LLM with randomly sampled parent solutions from the previous generation, followed by a round of tournament sampling to refine the set of parents to sample from in the next iteration based on their objective scores. Similar to BOPRO, BLMX can be constructed by modifying the tournament selection to instead leverage the cosine similarity scores with respect to the BO proposal instead of the black-box function scores.

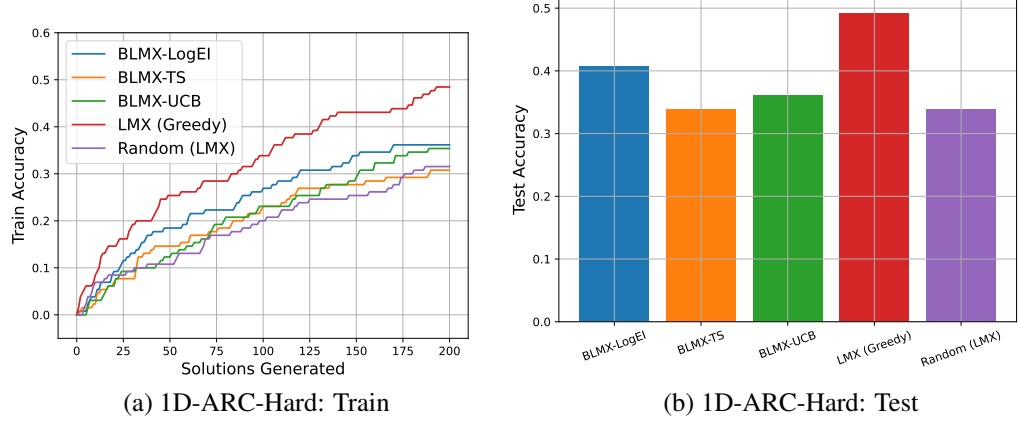

(a) 1D-ARC-Hard: Train
(b) 1D-ARC-Hard: Test

Figure 10: **LMX Performance on 1D-ARC.** Performance on 1D-ARC-Hard using LMX-decoding (instead of OPRO) averaged over 130 problem instances on a single run. Overall, the trends remains the same as with OPRO-decoding, but all methods show a drop in performance.

**Results on 1D-ARC-Hard.** To implement the BLMX strategy, we first double the number of top-$k$ candidates as compared to those used with BOPRO, followed by randomly selecting $k$ of them to

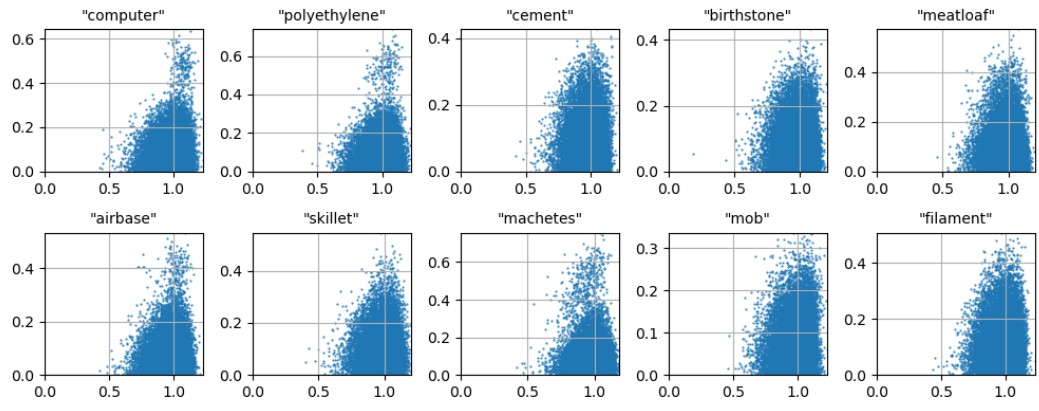

Figure 11: **Diagnostic for representation quality on Semantle.** The scatter plot shows the distance between candidate solutions of 10 hidden words in Semantle in the representation space and their corresponding difference in terms of their black-box scores. Useful representations for BO should result in no points in the top-left quadrant of the plots.

include. We, additionally, do not provide scores in the prompt and instruct the model to *combine* the included solutions to generate new ones. Figure 10 shows our results.

### A.5.3 VISUALIZING SEARCH TRAJECTORIES FOR 1D-ARC

We aim to visualize the exploration-exploitation ability in BOPRO by plotting the search trajectory using t-SNE contour plots of the latent space of embedding vectors. To do this, we aggregate all solutions generated across runs from each of our evaluated methods to construct a map of the effective search space for each task. In Fig. 5(c,d), we show search trajectories with BOPRO-TS and OPRO on a single problem instance, which demonstrates that BOPRO is indeed able to conduct effective exploration of the search space.

### A.5.4 DIAGNOSTIC REPRESENTATION PLOT: SEMANTLE

To contrast the diagnostic plot shown for 1D-ARC in Fig. 6, we show the same plot for Semantle in Fig. 11, a task where BOPRO does perform well. We clearly see that our desideratum from such a diagnostic plot of having low score differences when L2 distances are low are well satisfied.

### A.5.5 ABLATION: EFFECT OF NUMBER OF WARM-START EXAMPLES: SEMANTLE

In this section, we perform an ablation to evaluate the effect of the number of warm-start examples used to initialize the surrogate model. To do this, we perform an experiment on Semantle using 5 target words and 5 warm-start sets, resulting in 25 problem instances. We evaluate performance using $W = \{2, 5, 10, 20, 40\}$ warm-start examples. Our results are shown in Fig. 12. We find that performance initially increases with an increase in the number of warm-start examples from $W = 2$ till $W = 20$. At $W = 40$, we find that performance lowers. This result follows from the fact that while starting with more warm-start examples initializes the surrogate model with more accurate beliefs about the search space, there is a trade-off between using the available budget to perform an initial sampling vs. applying that budget to targeted search.

### A.5.6 ABLATION: EFFECT OF DIMENSIONALITY REDUCTION ON SEMANTLE

In Fig. 13, we show the effect on search performance of using PCA dimensionality reduction, reducing the original Qwen embeddings to 10 dimensions. We run an oracle experiment, where we replace the LLM latent-to-text decoding step with a selection procedure from a fixed set of 4000 words based on nearest-neighbor retrieval to the proposed BO vector. We do this to remove any errors from LLM sampling in our analysis. The results clearly show that using PCA-based dimen-

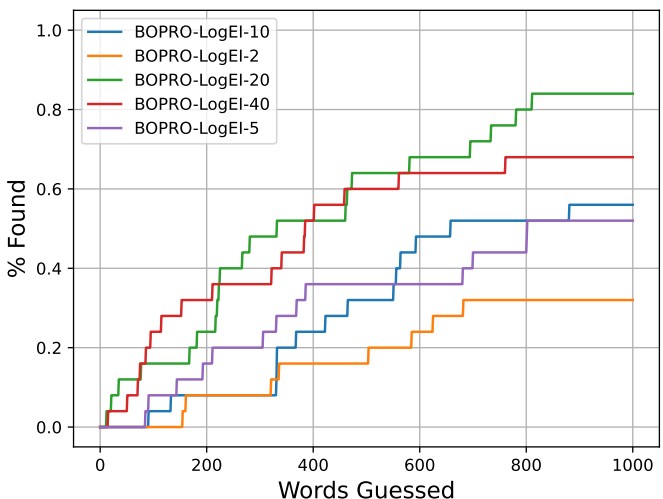

Figure 12: Effect of the number of warm-start examples on Semantle.

sionality reduction hampers search performance in BOPRO by a large amount (nearly 40 percentage points).

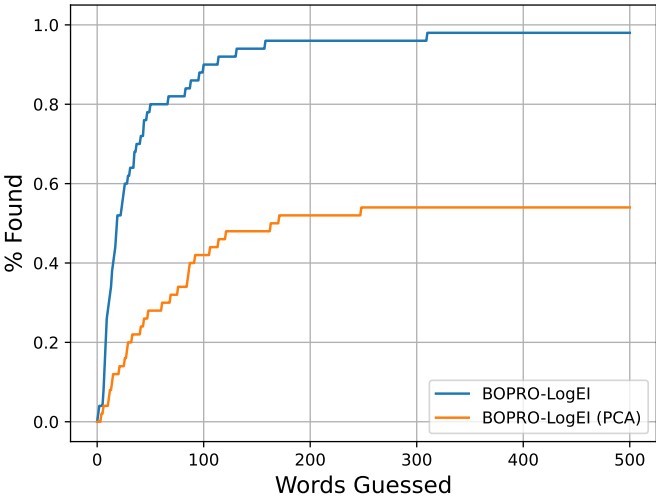

Figure 13: Effect of PCA using a selection-based BOPRO run over 4000 words on Semantle.

### A.6 LLM PROMPTS

#### A.6.1 WARM-START GENERATION

**Semantle**

```
Your task is to guess a hidden test word from the English dictionary.
Start by guessing {n_cands} diverse words to maximize your chance of
guessing the hidden word. Be creative.
```

```
Do not guess any of the following words: {list of words repeated by the
LLM in the previous iteration}.
```

**Dockstring**

```
Your task is to find the optimal drug molecule that has both a high
druglikeness (QED) as well as a strong binding affinity (vina) to the
target protein. While both properties are important, binding affinity is
twice as important as the druglikeness. Here is the target protein: {
target_protein}.

Do not propose any of the following molecules: {list of molecules
repeated by the LLM in the previous iteration}.
```

**1D-ARC**

```
Given the following {len(task_demos)} examples of input-output grids of
integers from 0-9, your task is to determine the transformation logic
common to all examples and provide a python function that converts each
input grid into its corresponding output grid. Here are the examples:

INPUT #1:
[[1,1,1,1.....]]
OUTPUT #1:
[[1,1,1,1.....]]
...

Each number in a grid may be treated as a unique color and a 0 may be
treated as an empty or black cell. Continuous sequences of numbers row-
wise, column-wise, or diagonally may represent objects forming patterns.
The transformation involves figuring out how these object patterns change
 from the input to the output. For e.g., object patterns may contain
shapes like rectangles, triangles, or crosses which can then be mirrored,
 rotated, translated, deformed, combined, repeated, etc.

Your functions should have the following signature:
```python
def transform(input: np.ndarray) -> np.ndarray:
    # Your code here
```
The following packages are already imported so do not repeat them: `numpy
 as np` and `itertools`

Start by guessing exactly {n_cands} diverse solutions that could solve
the task. Make sure to not repeat any previous docstring or code. Be
creative in the logic of your proposals!

Do not repeat any of the following guesses:

REPEAT #1:
<docstring>This function...</docstring>
```python
def transform(input: np.ndarray) -> np.ndarray:
    ...
```

REPEAT #2:
...

(Note: directly start your response using the specified output format)
```

### A.6.2  SOLUTION REPRESENTATION

To compute embeddings for the generated solutions, we first map the raw solution text to a template to provide additional context for sentence-embedding models when needed:

- **Semantle:** `"{x}"`
- **1D-ARC:** `"Algorithm:{x}"`
- **Dockstring:** `"Protein target:{target} Molecule Candidate:{x}"`

### A.6.3  SOLUTION GENERATION

**Semantle**

```
Your task is to guess a hidden test word from the English dictionary. Use
 the below series of your previous guesses (in increasing) order of their
 similarity to the hidden word in terms of their *meaning*) to make a new
 guess. Your new guess should not have been made before and should score
higher than your previous guesses. Analyze your previous guesses to
decide what word could reach a target score of {target_score:.4f}.

If you guess an invalid word (i.e., not in the dictionary or a repeat
guess), you will get no score, so stick to proper, single-word English
words, and do not repeat your previous guesses!

Now, guess exactly n=%s new word(s) that could give you a target score of
 {target_score:.4f}.

(Note: give only the word in the provided JSON format)
```

**Dockstring**

```
Your task is to find the optimal drug molecule that has both a high
druglikeness (QED) as well as a strong binding affinity (vina) to the
target protein. While both properties are important, binding affinity is
twice as important as the druglikeness. Here is the target protein: {
target_protein}. Use the below series of your previous guesses in
increasing order of their scalarized scores (between 0 and 1) to make a
new guess to maximize the score. Your new guess should not have been made
 before. Analyze your previous guesses to decide what molecule to guess
next. If you propose an invalid molecule or make a repeat guess, you will
 get no score, so stick to valid SMILES strings, and do not repeat your
previous guesses!

Now, guess exactly n=%s new molecule(s) that could be the optimal word.
Be creative!

Now, guess exactly n=%s new word(s) that could give you a target score of
 {target_score:.4f}.

(Note: give only a list of SMILES strings in the provided JSON format, e.
g. \
{{"response": ["SMILES1", "SMILES2", ...]}})
```

**1D-ARC**

```
Given the following {len(task_demos)} examples of input-output grids of
integers from 0-9, your task is to determine the transformation logic
common to all examples and provide a python function that converts each
input grid into its corresponding output grid. Here are the examples:

INPUT #1:
```

```
[[1,1,1,1.....]]
OUTPUT #1:
[[1,1,1,1.....]]
...

Each number in a grid may be treated as a unique color and a 0 may be
treated as an empty or black cell. Continuous sequences of numbers row-
wise, column-wise, or diagonally may represent objects forming patterns.
The transformation involves figuring out how these object patterns change
 from the input to the output. For e.g., object patterns may contain
shapes like rectangles, triangles, or crosses which can then be mirrored,
 rotated, translated, deformed, combined, repeated, etc.

Your functions should have the following signature:
```python
def transform(input: np.ndarray) -> np.ndarray:
    # Your code here
```
The following packages are already imported so do not repeat them: `numpy
 as np` and `itertools`

Here are your top previous guesses and scores (in ascending order of
accuracy compared to the target solution)

GUESS (score: ...):
<docstring> ... </docstring>

GUESS (score: ...):
<docstring> ... </docstring>

GUESS (score: ...):
<docstring> ... </docstring>

....

Analyze your previous guesses to make exactly {n} new guess(es) likely to
 give you a target score of {target_score:.4f}. Be creative in your logic
!

Do not repeat any of the following guesses:

REPEAT #1:
<docstring>This function...</docstring>
```python
def transform(input: np.ndarray) -> np.ndarray:
    ...
```

REPEAT #2:
...

(Note: directly start your response using the specified output format)
```

