# OpenReview forum: "Searching for Optimal Solutions with LLMs via Bayesian Optimization"
_ICLR.cc/2025/Conference — ICLR 2025 Poster_

### Official Review · Reviewer_pYWd · 2024-10-21

**Soundness:** 3
**Presentation:** 2
**Contribution:** 2
**Rating:** 6
**Confidence:** 3

**Summary:**

This paper introduces BOPRO, a novel Bayesian optimization-based search methodology for LLM-driven tasks, showing its exploratory capabilities and adaptability to task difficulty. Despite BOPRO’s strengths, it underperforms compared to a greedy baseline. The paper also identifies its limitations and suggests directions for future improvement.

**Strengths:**

- Combining the search capabilities of BO and the language capabilities of LLM is a research direction worth exploring.
- Plotting the search path (e.g., Fig.1 and Fig.5) is a great way to illustrate the trade-off between exploration and exploitation.
- The experiment conducted a detailed analysis of BOPRO underperforming.

**Weaknesses:**

- The claimed generality is not well demonstrated. The abstract states that a key issue in this field is the lack of "a one-size-fits-all approach" (line 14), and the introduction provides several examples that highlight the importance of a general search strategy. I initially believed that generality would be a major highlight of the proposed algorithm. However, it is disappointing to see that the experiments were conducted on only "two representative language search tasks" (line 110), which clearly does not showcase BOPRO as a general method. As shown in Appendix A.6, the prompts are specifically tailored to the given tasks and are not generalizable.
- The innovation is questionable. Bayesian optimization has proven its utility and been extensively explored due to its strong performance and wide applicability. If this paper simply applies LLMs to the BO framework in the context of two specific tasks, the novelty is doubtful. There are already numerous search algorithms combining LLMs with BO.
- The experiments are insufficient. The experiments only compare BOPRO with three basic search strategies: greedy, repeated sampling, and random. These methods are too simple to demonstrate the superiority of BOPRO. The author should include current state-of-the-art methods, such as those mentioned in [1-4], and conduct more comprehensive experiments with different background.
- The reproducibility is questionable. I intended to test the performance of the algorithm, but I could not find any code or materials to support reproducibility. For example, the author claims that BO is described in Appendix A.2, but the expressions for the acquisition functions and related parameters are not provided. Similarly, in the LLM section, there is no explanation for why the Sematle task uses the T5-base model while the 1D-ARC task uses the Qwen1.5 model. ICLR has clear expectations regarding reproducibility (https://iclr.cc/Conferences/2025/AuthorGuide), and the author is encouraged to include such information at the end of the paper. Perhaps a reproducibility statement is not necessary, but at least it should ensure that key parameters are covered in the paper.
- Expression. The entire manuscript should be carefully proofread; otherwise, it is not ready for publication. For instance, "warm-start" and "warmstart" are used interchangeably (lines 685 and 689).

[1] Joel Lehman, Jonathan Gordon, Shawn Jain, Kamal Ndousse, Cathy Yeh, and Kenneth O Stanley. Evolution through large models. In Handbook of Evolutionary Machine Learning, pp. 331–366. Springer, 2023.

[2] Chengrun Yang, Xuezhi Wang, Yifeng Lu, Hanxiao Liu, Quoc V Le, Denny Zhou, and Xinyun Chen. Large language models as optimizers. In The Twelfth International Conference on Learning Representations, 2024.

[3] Lichang Chen, Jiuhai Chen, Tom Goldstein, Heng Huang, and Tianyi Zhou. Instructzero: Efficient instruction optimization for black-box large language models. In Forty-first International Conference on Machine Learning, 2024.

[4] Krista Opsahl-Ong, Michael J Ryan, Josh Purtell, David Broman, Christopher Potts, Matei Zaharia, and Omar Khattab. Optimizing instructions and demonstrations for multi-stage language model programs. arXiv preprint arXiv:2406.11695, 2024.

**Questions:**

1. Why doesn't the author provide an overall framework diagram of the BOPRO or use the "Algorithm" environment in LaTeX to fully describe the BOPRO process? The current description is quite unclear, making it difficult for readers to form a comprehensive understanding of the relationship between BO and LLM.
2. Why are Qwen-1.5 and T5-base used as the base models? Specifically, (a) why are different models used for the two tasks? (b) why are other models (such as Llama) not evaluated, or why hasn't an ablation study been conducted on the base models? (c) If BOPRO is based on prompts, why not consider using the GPT-4 series as the base model (especially since the original paper on the 1D-ARC dataset also evaluated performance on GPT-4)?
3. As mentioned under "weaknesses", if the goal is to show the generality, comprehensive experiments on multiple search tasks are needed. The following problems could serve as useful references: combinatorial optimization (e.g., traveling salesman problem, knapsack problem), path planning and navigation, game problems (e.g., in reinforcement learning environments, Go, or chess), and program synthesis. While it may be challenging for the author to cover all types of tasks, covering too few problem scenarios is insufficient to demonstrate the generality of BOPRO. Why hasn't the author attempted more tasks?
4. The comparison methods are not strong enough; simple greedy strategies represent very basic approaches. In the introduction, the author lists a series of highly relevant methods and highlights their shortcomings. Why doesn't the author conduct comparative experiments to demonstrate these shortcomings (as mentioned in lines 51-52, "too expensive, not performant enough, or impractical in offering a general solution for search")?
5. Regarding exploration and exploitation, the author mentions that "we are interested in answering whether search with large language models (LLMs) can handle this trade-off"(line 43). Although the trajectory demonstrates a good balance, is this really the capability of the LLM? Most BO-based frameworks specifically balance exploration and exploitation, typically related to the choice of acquisition function. The BO acquisition functions used in the experiments all include sufficient exploration components, so is this balance due to them? This question is not well addressed and may require ablation studies to clarify.
6. The author mentioned that "BOPRO trails a strong greedy baseline in aggregate"(line 26-27). Is this due to the prompt design? Can better performance be achieved by modifying the prompt?

---

> ### Author Response · Authors · 2024-11-23
>
> > `The innovation is questionable. Bayesian optimization has proven its utility and been extensively explored due to its strong performance and wide applicability. If this paper simply applies LLMs to the BO framework in the context of two specific tasks, the novelty is doubtful. There are already numerous search algorithms combining LLMs with BO.`
> > `The experiments are insufficient. The experiments only compare BOPRO with three basic search strategies: greedy, repeated sampling, and random. These methods are too simple to demonstrate the superiority of BOPRO. The author should include current state-of-the-art methods, such as those mentioned in [1-4], and conduct more comprehensive experiments with different background.`
>
> Thank you for pointing out these issues. First, we have added the following additional experiments to address your concerns:
> - A new multi-objective task of molecule optimization to synthesize molecules with high QED and low Vina scores for 58 protein binding sites (please see Section 7.2).
> - A comparison with InstructZero (mentioned in [3]) on Semantle, which shows that InstructZero is only able to achieve 8% optimization accuracy, as compared to 66.67% achieved by the lowest-performing BOPRO variant (please see Table 1).
>
> Additionally, we clarify that:
> - Our primary baseline, earlier labeled “Greedy”, is in fact OPRO, which is referenced in [2]. We have updated the label on this baseline throughout the paper for clarity.
> - Our experiments with LMX and BLMX on 1D-ARC (please see Appendix A.5.2) are representative of the evolutionary algorithm strategy described in [1].
> - Finally we argue that the setting in [4] is artificial/easier than our setting as it performs BO over a fixed set of pre-generated candidates whereas in our setting, we generate new candidates, balancing exploration-exploitation based on the evolving uncertainty estimates of the solution space.
>
> Below, we also provide a **summary of our new results**:
> On Semantle (Section 7.1), we show that all BOPRO variants outperform OPRO by ≥10 points, whereas OPRO shows initial rapid gains but plateaus, getting stuck in a local optima. On the new task of molecule optimization (Section 7.2), we show that BOPRO produces better quality molecules that optimize both objectives (unlike OPRO) and 17% more valid proposals than OPRO, which tends to produce increasingly longer sequences causing the black-box function to fail. OPRO, thus, only finishes optimization for 12 of the 58 protein targets we evaluate on, whereas all BOPRO variants complete 100% of the tasks in the same wall-clock time.
>
> > `The claimed generality is not well demonstrated. The abstract states that a key issue in this field is the lack of "a one-size-fits-all approach" (line 14), and the introduction provides several examples that highlight the importance of a general search strategy. I initially believed that generality would be a major highlight of the proposed algorithm. However, it is disappointing to see that the experiments were conducted on only "two representative language search tasks" (line 110), which clearly does not showcase BOPRO as a general method. As shown in Appendix A.6, the prompts are specifically tailored to the given tasks and are not generalizable.`
>
> We clarify that the lack of generality we highlight in current systems is in the search strategy they employ, more specifically - their inability to dynamically determine when to explore and when to exploit, which requires systems to pre-define a strategy using heuristics. In contrast, we have shown that BOPRO, through Bayesian optimization, can dynamically update its behavior during search based on its evolving beliefs about the solution space as more evidence is collected.
>
> In terms of generality with respect to tasks, our experiments show that BOPRO is a strong method as long as the solution can be represented as a sequence of tokens and there is access to a good solution embedding function. To show this, we have added additional experiments (Section 7.2) on a new task of molecule optimization, and find that BOPRO outperforms the baselines. Thank you for pointing this out!
>
> Indeed, the instruction prompts that we use for different tasks are different and simply specify the task description to the LLM. Our contribution is not in the specific instruction prompts used, but in the search framework that combines latent-space BO with LLM-based decoding, which remains constant across tasks.

---

> > ### Author Response · Authors · 2024-11-23
> >
> > > `The reproducibility is questionable. I intended to test the performance of the algorithm, but I could not find any code or materials to support reproducibility. For example, the author claims that BO is described in Appendix A.2... there is no explanation for why the Sematle task uses the T5-base model while the 1D-ARC task uses the Qwen1.5 model. ... Perhaps a reproducibility statement is not necessary, but at least it should ensure that key parameters are covered in the paper.`
> >
> > We apologize for this missing information in the previous draft! We have added an extended method details section in Appendix A.2 and all our experimental settings in Appendix A.4. Additionally, we emphasize that we commit to open-sourcing all our code.
> >
> > Thank you for pointing out the discrepancy in the choice of embedding functions used in our previous experimental setting . We have now unified our setup and use GTE-Qwen2-1.5b-Instruct for both Semantle and 1D-ARC. Our new results for Semantle are shown in Section 7.1. Note that apart from a difference in embedding function, these new experiments no longer use dimensionality reduction and were run for 1000 evaluations (earlier: PCA dimensionality reduction was used for 500 evaluations). Our results in Fig.2(a) show that all variants of BOPRO outperform all baselines.
> >
> > > `Why are Qwen-1.5 and T5-base used as the base models? Specifically, (a) why are different models used for the two tasks? (b) why are other models (such as Llama) not evaluated, or why hasn't an ablation study been conducted on the base models? (c) If BOPRO is based on prompts, why not consider using the GPT-4 series as the base model (especially since the original paper on the 1D-ARC dataset also evaluated performance on GPT-4)?`
> >
> > Thank you for these questions. We have updated our experiments to unify the setup and use GTE-Qwen-2-1.5b-Instruct as the embedding model for both Semantle and 1D-ARC (Section 7.1 and 7.3). Our choice for this model is motivated by its pre-training data containing both natural language text and code, both of which are relevant to the tasks at hand. For our new experiment on molecule optimization, we instead use Molformer, an embedding model pre-trained on SMILES strings (Section 7.2).
> >
> > With regard to different LLMs for solution generation, we have added the following new experiments in addition to our main results with Mistral-Large-2407:
> > - Gemma-2-2b-It experiment on Semantle (Appendix A.5.1; Fig. 9(a))
> > - GPT-4o experiment on Semantle (Appendix A.5.1; Fig. 9(b)). Please note that we were only able to evaluate one BOPRO variant and OPRO due to budget constraints.
> > - LLaMA-3.1-8b-Instruct experiment comparing BOPRO with InstructZero (Section 7.4; Table 1)
> >
> > Our results show BOPRO outperforming the baselines similar to our main results, albeit with different amounts of gaps in some cases. Please also note that a direct comparison with the original paper [1] is addressed by our observation that simple repeated sampling with Mistral-Large achieves a higher accuracy of 81.48% than their reported accuracy of 73.1% (see L399-405).
> >
> > [1] Hypothesis search: Inductive reasoning with language models. (Wang et al., 2023)
> >
> > > `As mentioned under "weaknesses", if the goal is to show the generality, comprehensive experiments on multiple search tasks are needed. The following problems could serve as useful references: combinatorial optimization (e.g., traveling salesman problem, knapsack problem), path planning and navigation, game problems (e.g., in reinforcement learning environments, Go, or chess), and program synthesis. While it may be challenging for the author to cover all types of tasks, covering too few problem scenarios is insufficient to demonstrate the generality of BOPRO. Why hasn't the author attempted more tasks?`
> >
> > Thank you for these suggestions! To address this concern, we have now added a challenging molecule optimization task (Section 7.2), where the goal is to synthesize new molecules that jointly optimize both QED (druglikeness) scores as well as binding affinity (Vina scores) to a target protein. Our evaluation is done over 58 protein binding sites from the Dockstring benchmark, and our results show that BOPRO outperforms all the baselines. We also see in Fig. 3 that BOPRO is able to propose molecules that indeed jointly optimize both objectives whereas OPRO fails to do so. In fact, in the same wall-clock time of 2 days, OPRO only finishes optimization for 12 of the 58 proteins, whereas all BOPRO variants complete 100% of the tasks, producing higher quality molecules. Also, please note that program synthesis is already part of our 1D-ARC evaluation, results for which are presented in Section 7.3.
> >
> > Given the limited window in the rebuttal period, we are unable to present additional experiments on combinatorial optimization and path planning. But these would be great additions that we can incorporate in the final paper.

---

> > > ### Author Response · Authors · 2024-11-23
> > >
> > > > `The comparison methods are not strong enough; simple greedy strategies represent very basic approaches. In the introduction, the author lists a series of highly relevant methods and highlights their shortcomings. Why doesn't the author conduct comparative experiments to demonstrate these shortcomings (as mentioned in lines 51-52, "too expensive, not performant enough, or impractical in offering a general solution for search")?`
> > >
> > > We summarize our response from W2, where we address each of the baseline references you provided.
> > >
> > > First, we clarify that:
> > > - Our primary baseline, earlier labeled “Greedy”, is in fact OPRO, which is referenced in [2]. We have updated the label on this baseline throughout the paper for clarity.
> > > - Our experiments with LMX and BLMX on 1D-ARC (please see Appendix A.5.2) are representative of the evolutionary algorithm strategy described in [1].
> > > - Finally we believe that the setting in [4] is artificial/easier than our setting as it performs BO over a fixed set of pre-generated candidates whereas in our setting, we generate new candidates, balancing exploration-exploitation based on the evolving uncertainty estimates of the solution space.
> > >
> > > Second, we have added the following new experiments to address the remaining concerns:
> > > - A new multi-objective task of molecule optimization to synthesize molecules with high QED and low Vina scores for 58 protein binding sites (please see Section 7.2).
> > > - A comparison with InstructZero (mentioned in [3]) on Semantle, which shows that InstructZero is only able to achieve 8% optimization accuracy, as compared to 66.67% achieved by the lowest-performing BOPRO variant (please see Table 1).
> > >
> > > Please let us know if these address your concerns!
> > >
> > > > `Regarding exploration and exploitation, the author mentions that "we are interested in answering whether search with large language models (LLMs) can handle this trade-off"(line 43). Although the trajectory demonstrates a good balance, is this really the capability of the LLM? Most BO-based frameworks specifically balance exploration and exploitation ... This question is not well addressed and may require ablation studies to clarify.`
> > >
> > > Our intention for the referenced sentence was to specify our research question, which says: can we design a search method for LLMs that is able to balance exploration and exploitation? Our solution to the problem indeed uses BO’s ability to balance exploration and exploitation and we show, via our proposed method BOPRO, that it is possible to design a method that can leverage this signal to guide generation with LLMs.
> > >
> > > We have updated the line in the paper to the following to make this intention more clear: “we are interested in answering whether a search **method** with large language models (LLMs) can be designed to handle this trade-off automatically.”
> > >
> > > > `The author mentioned that "BOPRO trails a strong greedy baseline in aggregate"(line 26-27). Is this due to the prompt design? Can better performance be achieved by modifying the prompt?`
> > >
> > > Our new experimental setting for Semantle runs BOPRO without dimensionality reduction and for double the number of iterations as before, resulting in BOPRO outperforming all baselines, including our main baseline of OPRO (greedy) (Section 7.1). BOPRO also outperforms all baselines (including OPRO (greedy)) on our new task of molecule optimization (Section 7.2). Finally, described in Section 7.3 and analyzed in Section 8, BOPRO does still trail OPRO on program search due to a lack of fine-grained code representations.
> > >
> > > Note that the instruction prompt used across all the methods in our study is exactly the same. The only part of the prompt that changes between methods are the in-context examples, which for BOPRO is selected using BO, for OPRO is selected greedily, for random by random sampling, and for repeated sampling we use no examples (only the task instruction). Therefore, any modification (good or bad) to the instruction prompt should provide similar increase or decrease for all methods. Our current instructions prompts were manually written by running small experiments that verified stability in producing output in the correct format and low repetitions.
> > >
> > > > `Why doesn't the author provide an overall framework diagram of the BOPRO or use the "Algorithm" environment in LaTeX to fully describe the BOPRO process? The current description is quite unclear, making it difficult for readers to form a comprehensive understanding of the relationship between BO and LLM.`
> > >
> > > Thank you for pointing this out! We have now added an overview figure for BOPRO as Figure 1 in the revision. We hope this addresses your concern.
> > >
> > > > `Expression. The entire manuscript should be carefully proofread; otherwise, it is not ready for publication. For instance, "warm-start" and "warmstart" are used interchangeably (lines 685 and 689).`
> > >
> > > Thank you for pointing out the typos! We have addressed them in the new revision and done another round of proof-reading.

---

> > > > ### Comment · Reviewer_pYWd · 2024-11-25
> > > >
> > > > Thanks for the author's response! I noticed that the authors made substantial changes to the paper, adding experiments and responding to my concerns.
> > > >
> > > > I generally agree with the changes the authors made:
> > > >
> > > > 1. The authors provide more detailed descriptions of reproducibility and modify unfair experimental settings. This is necessary.
> > > > 2. The authors enhance the performance of BOPRO. The authors also add experiments on different tasks. This is aligned with the motivation.
> > > > 3. The authors conduct experiments on more LLMs (such as GPT-4o), which are also necessary.
> > > >
> > > > Some questions:
> > > >
> > > > 1. In my view, BOPRO has the potential to solve more complex problems such as solving complex string optimization problems or tasks in motion control scenarios. BOPRO is both worthwhile and should be explored further. While I preliminarily acknowledge the experiments in this paper, I still suggest the authors try more problem scenarios, especially more complex tasks, to demonstrate the superior performance of BOPRO.
> > > > 2. I have preliminarily experimented with using LLMs to guide BO in exploration. One key issue is that LLMs are better at handling textual information than numerical data. This means that when the dimensionality of the objective function increases (i.e., x becomes a high-dimensional vector, such as 200 dimensions), LLMs may not perform well in optimization tasks, as such tasks often require exact numerical values, which exacerbates the high-dimensional bottleneck problem that BO already faces. (This is also why I believe LLMs are more suitable for string optimization problems.) How do the authors view this issue?
> > > > 3. Another limitation of LLM+BO is that when the prompt for the LLM becomes too long, the LLM might struggle to focus attention, manifesting as a lack of understanding of the problem requirements or failing to respond in the specified format. This is one of the reasons limiting LLMs in handling large-scale optimization problems. The use of LLMs in this paper relies on prompt design, and how to design a reasonable prompt directly impacts performance. However, it seems that the prompts in this paper are manually specified by the authors based on specific tasks. How do the authors view this issue?
> > > > 4. BO is a sequential optimization process that relies on historical information. Can it be integrated with certain scenarios of LLMs, such as multi-turn conversations with LLMs?
> > > > 5. During the optimization process, is it possible to enable LLMs to gradually gain awareness of domain knowledge and problem characteristics, thereby better guiding the optimization process? (This is just a discussion with you.)
> > > >
> > > > I would like to once again thank the authors for their efforts in addressing my concerns! I have increased my score accordingly!

---

> > > > > ### Author Response · Authors · 2024-11-25
> > > > >
> > > > > We’re very glad our changes have addressed some of your concerns! We hope the following can help address the remainder.
> > > > >
> > > > > > `…BOPRO has the potential to solve more complex problems such as solving complex string optimization problems or tasks in motion control scenarios. BOPRO is both worthwhile and should be explored further. … I still suggest the authors try more problem scenarios, especially more complex tasks, to demonstrate the superior performance of BOPRO.`
> > > > >
> > > > > We agree completely that BOPRO could be applied to a large number of other tasks. We believe that the tasks of word optimization, molecule optimization, and hypothesis+program optimization would fall under the categorization of string optimization problems. As shown by our baselines and reference to the low performance in prior works, these tasks, particularly molecule optimization and 1D-ARC, are quite challenging.
> > > > >
> > > > > Motion planning and combinatorial optimization would indeed be very interesting test-beds for BOPRO. Given the limited time during rebuttal and the expense involved in sequential optimization experiments, we are currently unable to provide results on these, unfortunately. We commit to adding at least one of these as new experiments in the final paper.
> > > > >
> > > > > > `I have preliminarily experimented with using LLMs to guide BO in exploration. One key issue is that LLMs are better at handling textual information than numerical data. This means that when the dimensionality of the objective function increases (i.e., x becomes a high-dimensional vector, such as 200 dimensions), LLMs may not perform well in optimization tasks, as such tasks often require exact numerical values, which exacerbates the high-dimensional bottleneck problem that BO already faces. (This is also why I believe LLMs are more suitable for string optimization problems.) How do the authors view this issue?`
> > > > >
> > > > > When operating with numerical data, such as in hyperparameter optimization, we do think BOPRO could provide a path to a solution in the future (even with hundreds of parameters embedded as text; program solutions in 1D-ARC can be much longer than that in token length) but is likely to currently run into the same issue as that observed on 1D-ARC, i.e., candidate solutions will have very low edit-distances and sequence embeddings will not be able to tell them apart, thus being unable to provide useful signal for posterior updates. For this reason, we strongly believe that a fruitful future investigation would be to train fine-grained embedding models that are able to model the nuanced differences between textual sequences, particularly in domains with code and numerics, to guide the optimization procedure in BOPRO.
> > > > >
> > > > > > `Another limitation of LLM+BO is that when the prompt for the LLM becomes too long, the LLM might struggle to focus attention, manifesting as a lack of understanding of the problem requirements or failing to respond in the specified format. This is one of the reasons limiting LLMs in handling large-scale optimization problems.`
> > > > >
> > > > > Indeed, providing the full optimization trajectory in a prompt is either infeasible when operating with smaller LLMs or not effective even when using LLMs with larger context windows. For this reason, our design decision in BOPRO was to operate in a latent embedding space, such that the heavy-lifting of deciding where to explore and exploit next is provided by a dedicated Gaussian process model and the LLM is only used as a decoder to convert a proposed latent vector back into text. This latter process in BOPRO only ever leverages a handful of examples to provide the textual gradient for decoding that vector, and so does not suffer from issues from limited context-window size.
> > > > >
> > > > > Having said that, we have acknowledged in the paper that our decoding procedure does not result in a new solution sequence that has the precise embedding as the proposed BO vector. Our experiments have shown, however, that it allows us to move towards the optimal solution during search. To make this decoding procedure more precise and alleviate the need to do prompting, we do think that a promising approach is to train a vec2text model such as in [1, 2].
> > > > >
> > > > > [1] Text Embeddings Reveal (Almost) As Much As Text (Morris et al., 2023)
> > > > > [2] vec2text with Round-Trip Translations (Cideron et al., 2022)

---

> > > > > > ### Author Response · Authors · 2024-11-25
> > > > > >
> > > > > > > `The use of LLMs in this paper relies on prompt design, and how to design a reasonable prompt directly impacts performance. However, it seems that the prompts in this paper are manually specified by the authors based on specific tasks. How do the authors view this issue?`
> > > > > >
> > > > > > We re-emphasize that the main differences between our prompts for different tasks is only in the task descriptions, e.g., whether to generate words, SMILES strings, or hypotheses/programs, and the format which we want the LLM to generate solutions in (please see Appendix A.6.3 for the prompts we use). Importantly, the way the in-context examples are selected and ordered using BOPRO, which is the core part of the method, remains unchanged across tasks. Please note that this is the dominant paradigm when using the in-context learning ability of LLMs for any task.
> > > > > >
> > > > > > > `​​BO is a sequential optimization process that relies on historical information. Can it be integrated with certain scenarios of LLMs, such as multi-turn conversations with LLMs?`
> > > > > >
> > > > > > This is a fascinating question! Our motivation and focus in the present work was to take the first step towards combining BO and LLMs in a performant and practical procedure for sequence optimization at test-time in a _single-state_ setup. By single-state, we mean a decision-process where the goal is to optimize only one decision given a fixed context in which the decision is taken, as compared to a multi-step decision-making process, where taking one decision results in the agent reaching a new context (possibly with a new set of actions available to it). We think it could be feasible to apply our procedure as-is in multi-turn (multi-state) conversations, where it is first determined that a user’s query requires search/optimization, followed by using BOPRO with either a ground-truth verifier or a learned reward model to optimize candidate responses within a reasonable search budget. Note that this is akin to the recent OpenAI o1 model for which we, unfortunately, do not have many technical details publicly available. This could be excellent future work to explore for BOPRO.
> > > > > >
> > > > > > > `During the optimization process, is it possible to enable LLMs to gradually gain awareness of domain knowledge and problem characteristics, thereby better guiding the optimization process? (This is just a discussion with you.)`
> > > > > >
> > > > > > We first note that the surrogate modeling in BOPRO does attempt to gain this awareness of domain knowledge and problem characteristics in the latent embedding space. If we understand correctly, your question hints at making the evolving beliefs in this latent-space more explicit, perhaps as textual sequences. That could be interesting. One interpretation of this suggestion is to augment the search space that BOPRO operates over from only optimizing the solution sequence to also optimizing over features/characteristics of the solution. Another possibility, and this is perhaps an advantage of using a prompting-based approach to decode sequences, is that latent BO proposals in each BOPRO iteration could be used to elicit such domain knowledge and problem characteristics by simply modifying the decoding prompt to aggregate the common characteristics described by the BOPRO-selected in-context examples, followed by solution generation.
> > > > > >
> > > > > > ---
> > > > > >
> > > > > > Thank you, again, for engaging in a constructive dialog! Please let us know if our answers address your remaining concerns. We are happy to provide additional discussion on any aspect or answer any additional questions. If you think fit, we request that you consider further raising your score.

---

> > > > > > > ### Comment · Reviewer_pYWd · 2024-11-26
> > > > > > >
> > > > > > > Thank you for your reply! I am glad to participate in the discussion with you. I have improved my score again!
> > > > > > >
> > > > > > > A small suggestion: your extensive revisions to the manuscript have greatly improved its quality, but may also introduce problems  such as typos and grammar. I suggest you check the entire paper carefully before publishing.

---

> > > > > > > > ### Author Response · Authors · 2024-11-26
> > > > > > > >
> > > > > > > > We really appreciate getting your positive feedback on the revised manuscript; thank you for your new score! We will make sure to go through multiple rounds of proof-reading before submitting a camera-ready version.

---

### Official Review · Reviewer_dfPF · 2024-10-27

**Soundness:** 3
**Presentation:** 4
**Contribution:** 2
**Rating:** 8
**Confidence:** 3

**Summary:**

This paper introduces BOPRO, a novel search method designed to improve the identification of optimal solutions by combining Bayesian Optimization with an LLM-based search strategy. BOPRO operates through an iterative search process, in which Bayesian Optimization is applied over a latent-space embedding derived from the LLM-generated prompts.

An empirical evaluation of BOPRO is performed on two benchmark tasks for language-based search, wherein it is found that BOPRO is outperformed by a greedy baseline approach.

Further investigation of the search behaviour of BOPRO is performed, revealing that it has strong exploration capabilities and can adapt to problem difficulty, but that it fails to effectively exploit promising regions of the search space.

**Strengths:**

The paper is well-written and relatively easy to follow. The proposed method has many details and "moving parts" but these are clearly outlined and their significance explained. The same goes for the experimental setup and results.

The paper is open and forthright about the limitations of BOPRO.

The paper includes nice big figures that are intuitive and easy to understand.

Despite the essentially negative results, the paper does a good job of clearly outlining the experimental investigation of BOPRO.

Given the popularity of LLMs and the effectiveness of Bayesian Optimization, I think the investigation described in this paper can be seen as a valuable contribution and could be a helpful starting point for future research.

**Weaknesses:**

The results are essentially negative (i.e., BOPRO is beaten by the greedy baseline and/or repeated sampling), which must be considered a weakness.

Figure 2: It would be easier to read the plots if they indicated somehow which lines were the three BOPRO variants (e.g., with different line styles and/or legend labels) and which were the baselines.

I think the related work would be nice to see earlier in the paper, particularly the part about other ways that Bayesian Optimization and LLMs have been combined.

Some minor issues not affecting my recommendation:

- Line 120 and elsewhere:  "For e.g.," this reads as too informal. Use "For example,".

- Figure 3 caption: There is maybe a word missing? "... random sampling with low due to ..."

- Section 7.2 Heading and elsewhere: "v/s" -> "vs."

**Questions:**

I wonder if the authors could explain the importance/significance of separating the solved from the unsolved cases in the various figures.

There seem to be a number of heuristics that were used. For example normalizing scores between 0.1 and 0.8, and the use PCA for dimensionality reduction (over other methods). It is stated that these improved performance. Was the performance improvement large? Is it expected that this will hold for other search tasks, or will these heuristics need to be adjusted in general?

---

> ### Author Response · Authors · 2024-11-23
>
> > `The results are essentially negative (i.e., BOPRO is beaten by the greedy baseline and/or repeated sampling), which must be considered a weakness.`
>
> Please see our new results for Semantle (Section 7.1), where we show that all BOPRO variants outperform OPRO by ≥10 points, whereas OPRO shows initial rapid gains but plateaus, getting stuck in a local optima. We have also added a new task of molecule optimization (Section 7.2), where we show that BOPRO produces better quality molecules that optimize both objectives (unlike OPRO) and 17% more valid proposals than OPRO, which tends to produce increasingly longer sequences causing the black-box function to fail. OPRO, thus, only finishes optimization for 12 of the 58 protein targets we evaluate on, whereas all BOPRO variants complete 100% of the tasks. Our new experimental setting runs BOPRO without dimensionality reduction and, in Semantle, for double the number of iterations as before. Our method does still trail OPRO on program search (Section 7.3), and we present a detailed analysis of this failure case in Section 8. First (Section 8.1), we show that BOPRO indeed is able to balance exploration and exploitation through both quantitative and qualitative analyses. Then (Section 8.2), we show that the issue lies in off-the-shelf code representations being unable to distinguish between sequences with low edit-distances. This is an important limitation in current code embedding models, which would be useful to address in future work.
>
> > `There seem to be a number of heuristics that were used. For example normalizing scores between 0.1 and 0.8, and the use PCA for dimensionality reduction (over other methods). It is stated that these improved performance. Was the performance improvement large? Is it expected that this will hold for other search tasks, or will these heuristics need to be adjusted in general?`
>
> Thank you for this question. Our new experimental setting addresses both of these concerns. First, we removed dimensionality reduction altogether, thus, reducing one additional choice a practitioner needs to make. This has also resulted in a significant boost for BOPRO, e.g., showing 10 percentage point gains over OPRO on Semantle. Our earlier setting of using PCA was informed by prior work and some initial results, which, in hindsight, we should have expanded to come to a final decision. Second, our new setting also removes scores from the prompt, which we found provide modest gains for all the methods (BOPRO and baselines), presumably because of the LLM’s inability to accurately interpret numbers associated with the solutions, indicating that the ordering of the examples is more important than their actual scores. We have, therefore, removed both considerations, simplifying the design of the method.
>
> > `I wonder if the authors could explain the importance/significance of separating the solved from the unsolved cases in the various figures.`
>
> Our new revision keeps only one example of the search trajectory comparison between BOPRO and OPRO, which shows how the ability to balance exploration and exploitation in BOPRO allows it to solve the task (Fig. 5(c)) whereas OPRO gets stuck in a local optima and fails to solve the task (Fig. 5(d)). We also note that this behavior can be quantitatively observed in our Semantle experiments (Fig. 2(a)), where OPRO shows rapid initial gains and then plateaus, whereas BOPRO shows steady gains and achieves the best performance. The additional comparisons we showed in our previous revision, simply showed additional examples of the same behavior as well as a contrasting case where a task was successfully solved by OPRO’s greedy strategy.
>
> > `Figure 2: It would be easier to read the plots if they indicated somehow which lines were the three BOPRO variants (e.g., with different line styles and/or legend labels) and which were the baselines.`
>
> Thank you for pointing this out! We have updated all our figures to clearly prefix all BOPRO methods with “BOPRO-”. Please let us know if this clears the confusion.
>
> > `I think the related work would be nice to see earlier in the paper, particularly the part about other ways that Bayesian Optimization and LLMs have been combined.`
>
> This is a great suggestion. We have made the change in the new revision!
>
> > `Some minor issues not affecting my recommendation:
> Line 120 and elsewhere: "For e.g.," this reads as too informal. Use "For example,".
> Figure 3 caption: There is maybe a word missing? "... random sampling with low due to ..."
> Section 7.2 Heading and elsewhere: "v/s" -> "vs."`
>
> Thank you! We have addressed these typos in the new revision.

---

> > ### Comment · Reviewer_dfPF · 2024-11-26
> >
> > I have read the authors' response and recognize that substantial improvements have been made. I remain positive about this paper and maintain my score.

---

> > > ### Author Response · Authors · 2024-11-26
> > >
> > > Thank you for acknowledging our new changes and your optimism about this line of work!

---

### Official Review · Reviewer_AwB4 · 2024-11-02

**Soundness:** 2
**Presentation:** 3
**Contribution:** 2
**Rating:** 5
**Confidence:** 3

**Summary:**

This work proposed a method leveraging LLM to perform optimization, named Bayesian-OPRO (BOPRO). This work can be seen as a generalization version of recent work OPRO. Their method is to intergrate bayesian optimization into LLM-based search, to be more specific, they use BO to determine the search direction for LLM by constructing a surrogate function on the latent space for value assignment. The experiment is performed in two tasks, however, the performance of this method underperforms OPRO in both tasks, although the authors have provided some in-depth analysis in terms of exploration capability and adaptbility.

**Strengths:**

1. The proposed method is novel, although there have been works focusing on combining LLM and BO to perform optimization tasks, the specific method of this work is different from other works. They perform BO process in latent space and design a reasonable encoding-decoding method.
2. The presentation is good, statements and visual presentation in the paper is clear.
3. Apart from main experiments, more in-depth analysis about adaptability and exploration are provided.

**Weaknesses:**

1. The experimental tasks in this works look easy and insufficient, which could not be enough for examine the performance.
2. The performance of this method is not good in the main experiment, as shown in Figure 2. Although the authors argue that this method could have stronger exploring capability and adaptability to different tasks, this kind of advantage isn't verified by the final performance which we should pay more attention to.

**Questions:**

1. The author mentioned in the introduction section that "Despite moderate success, these solutions are either too expensive, not performant enough, or impractical in offering a general solution for search.". How does this method resolve these issues? There seems no clear explanation of this aspect in the paper.
2. The authors mentioned in the Section 6 that " However, we find that our baseline of repeated sampling (RS) using a weaker Mistral-Large-2407 model is able to trivially surpass that score with an accuracy of 81.48%". I don't really get this statement, could the author please provide more detailed information of this? Using weaker model with less generations surpass the stronger model with more genrations go against my instincts.

---

> ### Author Response · Authors · 2024-11-23
>
> > `The experimental tasks in this works look easy and insufficient, which could not be enough for examine the performance.`
>
> To address your concern, we have added a new experimental setting of molecule optimization (Section 7.2) where the goal is to synthesize molecules with both high QED (druglikeness) scores and high binding affinity (low Vina scores). Our results show that BOPRO produces better quality molecules that optimize both objectives (unlike OPRO) and 17% more valid proposals than OPRO. OPRO further tends to produce increasingly longer sequences causing the black-box function to fail, resulting in finishing optimization for only 12 of the 58 protein targets we evaluate, whereas all BOPRO variants complete 100% of the tasks.
>
> Additionally, we note that our synthetic task of Semantle, albeit conceptually simple, represents a challenging test bed (also hosted as a popular online word game by the New York Times). The low scores with repeated sampling using LLMs as well as the fact that none of the compared methods perfectly solve the task underscores that this problem is non-trivial to solve.
>
> Lastly, our experimental setting for hypothesis+program search using 1D-ARC, as well as its original task ARC, is widely recognized as being challenging, which is evidenced by the low scores achieved by prior work (e.g., GPT-4 direct prompting achieves only 41.5% on the full 1D-ARC dataset [1]). Furthermore, we construct from the full dataset a hard subset of tasks that cannot be trivially solved by repeated temperature-based sampling in 100 iterations, resulting in our final test bed.
>
> [1] LLMs and the Abstraction and Reasoning Corpus: Successes, Failures, and the Importance of Object-based Representations (Xu et al., 2024)
>
>
> > `The performance of this method is not good in the main experiment, as shown in Figure 2. Although the authors argue that this method could have stronger exploring capability and adaptability to different tasks, this kind of advantage isn't verified by the final performance which we should pay more attention to.`
>
> Please see our new results for Semantle (Section 7.1), where we show that all BOPRO variants outperform OPRO by ≥10 points, whereas OPRO shows initial rapid gains but plateaus, getting stuck in a local optima. We have also added a new task of molecule optimization (Section 7.2), where we show that BOPRO produces better quality molecules that optimize both objectives (unlike OPRO) and 17% more valid proposals than OPRO, which tends to produce increasingly longer sequences causing the black-box function to fail. OPRO, thus, only finishes optimization for 12 of the 58 protein targets we evaluate on, whereas all BOPRO variants complete 100% of the tasks. Our new experimental setting runs BOPRO without dimensionality reduction and, in Semantle, for double the number of iterations as before. Our method does still trail OPRO on program search (Section 7.3), and we present a detailed analysis of this failure case in Section 8. First (Section 8.1), we show that BOPRO indeed is able to balance exploration and exploitation through both quantitative and qualitative analyses. Then (Section 8.2), we show that the issue lies in off-the-shelf code representations being unable to distinguish between sequences with low edit-distances. This is an important limitation in current code embedding models, which would be useful to address in future work.
>
> > `The authors mentioned in the Section 6 that " However, we find that our baseline of repeated sampling (RS) using a weaker Mistral-Large-2407 model is able to trivially surpass that score with an accuracy of 81.48%". I don't really get this statement, could the author please provide more detailed information of this? Using weaker model with less generations surpass the stronger model with more genrations go against my instincts.`
>
> Wang et al. [1], which is the work that we cite in Section 7.3, report achieving a test set accuracy of 73.1% on their provided subset. However, they do not compare to a simple repeated-sampling baseline as we have done. We conducted that experiment and found that even with a weaker model (Mistral-Large, not GPT-4), repeated sampling outperforms the technique used in Wang et al., which (a) suggests that their method may not be performing significant amounts of search, and (b) suggests that their chosen subset of questions may be an insufficient test bed for search. This is what motivated us to create a new hard subset of 1D-ARC composed of problems that were not solved in even 100 repeated samples with Mistral-Large. Note also that prior works have also found repeated sampling to be a strong method for code generation [2], therefore, the experimental setting in [1] should have included this important evaluation for completeness.
>
> [1] Hypothesis search: Inductive reasoning with language models. (Wang et al., 2023)
> [2] Large Language Monkeys: Scaling Inference Compute with Repeated Sampling (Brown et al., 2024)

---

> > ### Author Response · Authors · 2024-11-23
> >
> > > `The author mentioned in the introduction section that "Despite moderate success, these solutions are either too expensive, not performant enough, or impractical in offering a general solution for search.". How does this method resolve these issues? There seems no clear explanation of this aspect in the paper.`
> >
> > The methods we describe in this statement are OPRO (our main baseline) and LMX (an evolutionary algorithm, which we also present experiments with in Appendix A.5.2). Through our experiments, we have shown evidence that OPRO does trail our proposed method on 2 of the 3 tasks evaluated, where the 3rd task has a particular failure mode owing to poor representations, analysis for which we present in Section 8.2. In fact, on Semantle, we find that all variants of our BOPRO outperform OPRO by at least 10 percentage points. Therefore, the baselines are either not performant enough in the same budget, or require more expense (more iterations) to reach the performance of our method. E.g., in molecule optimization, we find that OPRO is only able to finish optimization for 12 of the 58 tasks in the same wall-clock time of 2 days as BOPRO, which finishes optimization for 100% of the tasks.
> >
> > As for “impractical in offering a general solution for search.”, our intention was to highlight that these method require making specific search-related design choices upfront and are incapable of dynamically adapting strategies as search progresses, something BOPRO is capable of doing (in Fig. 5(c), we show qualitatively that the search trajectory followed by BOPRO explores the search space and finds the optimal solution, whereas in Fig. 5(d), OPRO gets stuck in a local optima). We have edited this phrasing in our new revision to be more clear: “...or require fixing a search strategy through offline evaluations.” Thank you for pointing this out!

---

> > > ### Author Response · Authors · 2024-11-26
> > > **Did we address your concerns?**
> > >
> > > Dear reviewer AwB4, please let us know if our revision PDF (with details in the general comment) and our responses to your specific comments/questions address your concerns. We are more than happy to provide any more discussion and answer any additional questions. We hope these provide sufficient evidence to help you consider increasing your current score.

---

> > > > ### Comment · Reviewer_AwB4 · 2024-12-02
> > > >
> > > > I appreciate the authors' effort in addressing my concerns. The additional experiments and analyses looks good to me. I am willing to raise my score but will leave the decision to ACs.

---

> > > > > ### Author Response · Authors · 2024-12-02
> > > > >
> > > > > Thank you for acknowledging the changes! Please let us know what concerns potentially remain so that we can provide additional discussion for them.

---

### Official Review · Reviewer_mJwg · 2024-11-02

**Soundness:** 3
**Presentation:** 3
**Contribution:** 2
**Rating:** 6
**Confidence:** 3

**Summary:**

In this paper, the authors are interested in answering whether the LLMs can search for optimal solutions and reason in the planning problems. They extend the prior works which either attempt to predict the task difficulty to select the optimal solution or greedy sequential search, both of which are not always optimal. More specifically, the authors propose Bayesian-OPRO, that integrates Bayesian optimization with LLMs, which iteratively samples from new proposal distributions by prompting the LLM with a subset of its previous generations selected to explore different parts of the search space. They evaluated their method on two language-based search tasks.

**Strengths:**

Strengths:

1.	The paper is clear and well-written, with all the relevant works cited.

2.	The authors provide a thorough analysis of BOPRO’s exploration and exploitation capabilities and acknowledge that the proposed method fails to exploit.

3.	I like that this paper discusses the limitations and future work openly. These insights can help future researchers to build and improve upon the current methodology.

**Weaknesses:**

Weaknesses:

1.	The proposed methodology is dependent on the access to an external verifier.

2.	BOPRO is not able to balance exploration and exploitation and might not be ready for practical use of the reasoning in the LLMs.

3.	BOPRO might not be ready for practical use yet as it still consistently trails behind the greedy baseline.

**Questions:**

Questions:
1.	How does the number of warm-start prompts W influence the final performance of the model? Is there a specific range of W that balances exploration and exploitation effectively?

2.	In row 248, how do the authors sample a batch of candidates for each z’t+1? Could alternative sampling methods impact the diversity and quality of solutions generated?

3.	I am wondering if the choice of other dimensionality reduction technique can  help improve the performance of BOPRO?

4.	Do authors have any insights on what specific strategies can be introduced to improve BOPRO’s ability to balance exploration and exploitation?

---

> ### Author Response · Authors · 2024-11-23
>
> > `I am wondering if the choice of other dimensionality reduction technique can help improve the performance of BOPRO?`
> > `BOPRO might not be ready for practical use yet as it still consistently trails behind the greedy baseline.`
>
> Your intuition is precisely correct! Our analysis revealed that dimensionality reduction of embeddings was indeed a key reason for not seeing the expected results from BOPRO. Our new experiments, particularly on Semantle (Section 7.1), show that using the unreduced embeddings with BOPRO results in a significant improvement in performance, with BOPRO now outperforming all the baselines, including OPRO by ≥10 percentage points, whereas OPRO shows initial rapid gains but plateaus, getting stuck in a local optima (see Fig. 2(a)). This result is also interesting in the fact that we show that it is indeed possible to use Gaussian processes with very high dimensionality, contrary to a popularly-held belief from prior work and in the community. We have also added a new task of molecule optimization (Section 7.2), where we show that BOPRO produces better quality molecules that optimize both objectives (unlike OPRO) and 17% more valid proposals than OPRO, which tends to produce increasingly longer sequences causing the black-box function to fail. OPRO, thus, only finishes optimization for 12 of the 58 protein targets we evaluate on, whereas all BOPRO variants complete 100% of the tasks. Our method does still trail OPRO on program search (Section 7.3), and we present a detailed analysis of this failure case in Section 8, where we find that it is not because of an inability of BOPRO to balance exploration and exploitation (Section 8.1), but rather in the inability of off-the-shelf code embeddings in distinguishing between candidate code solutions with very low edit-distances (Section 8.2).
>
> > `How does the number of warm-start prompts W influence the final performance of the model? Is there a specific range of W that balances exploration and exploitation effectively?`
>
> Thank you for your thoughtful question! While a larger W allows for the initialization of the surrogate model with more accurate beliefs about the search space, the trade-off when generating warm-start examples (instead of selecting from a known set) is in the allocation of test-time compute in this initial sampling v/s using the budget to perform search. The answer to the optimal number of examples is, thus, closely tied to the task and the ability of a warmstart generator (e.g., repeated sampling with an LLM) to produce diverse samples. To show this, we have conducted an experiment on Semantle for 25 problem instances for 5 warmstart settings: W={2, 5, 10, 20 (original setting), 40}. Our results are shown in Appendix A.5.5, where we find that performance initially increases with increase in the number of warm-start examples until W=20, after which it dips a little. This is in-line with our described intuition above.
>
> > `In row 248, how do the authors sample a batch of candidates for each z’t+1? Could alternative sampling methods impact the diversity and quality of solutions generated?`
>
> Our procedure to sample candidate solutions given a BO proposal forms the second component of BOPRO, which we describe in Section 5.2 Latent-to-Text Decoding. In particular, given the BO proposal z’, we first select the most similar previously observed solutions using the cosine similarity between the embeddings of previous solutions and the BO proposal. We then order the selected observations in ascending order of similarity to construct the prompt (“search prompt”) that is input to an LLM. We then use temperature-based sampling to generate a batch of solutions with this search prompt. We believe that alternative sampling methods, such as greedy decoding or beam search, would be less suited to our method, given that our goal is not to sample one high-likelihood sequence, but to sample a batch of sequences from a target region in the search space, where temperature-based sampling can produce similar but distinct results.
>
> > `BOPRO is not able to balance exploration and exploitation and might not be ready for practical use of the reasoning in the LLMs.`
>
> In Section 8.1, we show evidence that BOPRO indeed is able to balance exploration and exploitation. Specifically, in Fig. 5(a,b), we show that even when starting with low warm-start scores, BOPRO is able to balance exploration and exploitation and achieve higher optimization accuracy than OPRO, which takes a greedy approach. Further, in Fig. 5(c,d), we show visually the search trajectories followed by BOPRO and OPRO. While OPRO gets stuck in a local optima and fails to solve the problem, BOPRO shows the ability to explore the search space and zero-in on the right solution.

---

> > ### Author Response · Authors · 2024-11-23
> >
> > > `The proposed methodology is dependent on the access to an external verifier.`
> >
> > The focus of our work is on black-box optimization, which, by definition, does indeed require the availability of an external verifier (the black-box function) [1,2,3]. Please also note that this assumption is not restrictive and can be applied to a large variety of problems. While the external verifiers can be replaced by learned task-specific verifiers such a process reward models [4], an advantage to using external verifier (e.g. python interpreter) is it allows us to avoid conflating results that could be introduced because of the errors made by the verifier.
> >
> > [1] Derivative-Free and Blackbox Optimization [Audet, 2017]
> > [2] Bayesian optimization (Garnett, 2023)
> > [3] A Tutorial on Bayesian Optimization (Frazier, 2018)
> > [4] Let's Verify Step by Step (Lightman et al., 2023)

---

> > > ### Comment · Reviewer_mJwg · 2024-11-25
> > >
> > > I thank the authors for their detailed response. It addresses all my concerns. Therefore, I am raising my score to 6.

---

> > > > ### Author Response · Authors · 2024-11-26
> > > >
> > > > We're very glad that we could address all of your concerns; thank you for raising your score! Please let us know what questions or discussions we could expand upon that may help you consider raising your scores further.

---

### Author Response · Authors · 2024-11-23
**Authors' General Response to Reviewers**

We sincerely thank all the reviewers for their time and expertise in evaluating our manuscript. We are very grateful for the positive feedback, including the recognition of BOPRO as a novel and useful contribution (reviewers AwB4, dfPF, pYWd), our clarity in writing and presentation (reviewers mJwg, AwB4, dfPF, pYWd), our analyses of the current limitations (reviewers mJwg, AwB4, pYWd), and BOPRO’s utility for future work to build upon (reviewers mJwg, dfPF).

The constructive feedback provided by the reviewers has been invaluable in improving the paper. Please find below the key changes we have made in our new revision in response to reviewer requests.

**New experiments:**
- Most significantly, two changes to our experimental setting have resulted in a dramatic improvement in performance: (1) removing dimensionality reduction, and (2) increasing the budget of evaluation for all methods. Our new results on Semantle (Section 7.1) show that BOPRO outperforms all the baselines, including our main baseline of OPRO by at least 10 percentage points. (Thank you reviewers mJwg and dfPF for constructive probing!)
- Next, we have added a challenging multi-objective search task of molecule optimization in Section 7.2, and show that BOPRO outperforms the baselines. Notably, (1) BOPRO demonstrates an ability to optimize both objectives jointly unlike OPRO (Fig. 3), and (2) OPRO only finishes optimization for 12 of 58 tasks, producing 17% fewer valid molecules than BOPRO. (In response to reviewers AwB4 and pYWd.)
- In light of our new results, we conduct a fresh analysis of the failure scenario with 1D-ARC in Section 8. In Section 8.1, we first show that BOPRO does indeed balance exploration and exploitation, and is not the cause of failure. In Section 8.2, we then show evidence (Fig. 6) to suggest that the key issue lies in candidate solutions with very low edit-distances and off-the-shelf code embeddings that cannot tell them apart.
- We have run a new experiment comparing InstructZero with BOPRO on Semantle in Section 7.4, showing superior results with BOPRO (requested by reviewer pYWd).
- We have run new experiments with additional LLMs, e.g., Gemma-2-2b-it, GPT-4o, and LLaMa-3.1-8b-instruct (Fig. 9 and Table 1, respectively), which show similar trends to our main results. (In response to reviewer pYWd.)
- We have added an experiment to show the effect of changing the number of warm-start examples in Appendix A.5.5. (requested by reviewer mJwg).
- We have additionally unified our experimental setup to use Qwen embeddings for both Semantle and 1D-ARC, while using Molformer for molecule optimization (in response to reviewer pYWd).
    - We provide an additional ablation experiment in Appendix A.5.6 to show the effect of using full-dimensional Qwen embeddings vs. PCA-reduced embeddings on search performance with BOPRO in an oracle setting.

**Writing changes:**
- We have added a new Figure 1 to provide a visual overview of BOPRO (requested by reviewer pYWd).
- We have moved the Related Works section up (requested by reviewer dfPF).
- We have provided all details on our experimental settings (including the BO setup and various hyperparameters) in Appendix A.4 (requested by reviewer pYWd).
- We have addressed all typos pointed out by reviewers. Thank you!
- We have made additional changes in different sections to support the new experiments and results:
  - Abstract and 9 Conclusion
    - Updated results, analysis, and added molecule optimization
  - 1 Introduction
    - Updated clarity of research question
    - Updated results and analysis description
    - Updated contributions
  - 2 Search Tasks
    - Added molecule optimization
  - 6 Experiments
    - Added InstructZero, LMX
    - Added molecule optimization
    - Added additional LLMs
  - 7 Results
    - Updated results
    - Added molecule optimization
    - Added InstructZero result
    - Added LMX result
  - 8 Why is BOPRO underperforming on program search?
    - Added new analyses

We would be very happy to address any further questions or concerns.

---

*Revision edit (2024/11/24; 7:05AM AOE): Corrected a typo in Fig. 2 caption.*

---

### Meta-Review · Area_Chair_HX3P · 2024-12-19

**Metareview:**

The paper presents a Bayesian optimization approach for in-context search with large language models. The overall approach is well formulated and justified with good results on two benchmarks. The paper also does a good job of analysis one negative result on code generation. I think the community will benefit from this paper and this paper has the potential of a multiple followup works. Therefore,  I recommend acceptance. I will strongly request the authors to make changes based on reviewers' suggestion especially related to open sourcing the code.

**Additional Comments On Reviewer Discussion:**

Reviewers had concerns about two baselines and the authors added one more task of molecule optimization where the proposed approach performs well. Reviewers noted missing implementation details and authors provided the details during the rebuttal period. Overall, all reviewers had positive views on the paper which weighed in my final decision.

---

### Decision · Program_Chairs · 2025-01-22

Accept (Poster)